# MICU1 controls cristae junction and spatially anchors mitochondrial Ca$^{2+}$ uniporter complex

Benjamin Gottschalk[1], Christiane Klec [1], Gerd Leitinger [2], Eva Bernhart[1], René Rost[1], Helmut Bischof [1], Corina T. Madreiter-Sokolowski[1], Snježana Radulović [1,2], Emrah Eroglu [1], Wolfgang Sattler[1,3], Markus Waldeck-Weiermair[1], Roland Malli [1,3] & Wolfgang F. Graier [1,3]

Recently identified core proteins (MICU1, MCU, EMRE) forming the mitochondrial Ca$^{2+}$ uniporter complex propelled investigations into its physiological workings. Here, we apply structured illumination microscopy to visualize and localize these proteins in living cells. Our data show that MICU1 localizes at the inner boundary membrane (IBM) due to electrostatic interaction of its polybasic domain. Moreover, this exclusive localization of MICU1 is important for the stability of cristae junctions (CJ), cytochrome c release and mitochondrial membrane potential. In contrast to MICU1, MCU and EMRE are homogeneously distributed at the inner mitochondrial membrane under resting conditions. However, upon Ca$^{2+}$ elevation MCU and EMRE dynamically accumulate at the IBM in a MICU1-dependent manner. Eventually, our findings unveil an essential function of MICU1 in CJ stabilization and provide mechanistic insights of how sophistically MICU1 controls the MCU-Complex while maintaining the structural mitochondrial membrane framework.

[1] Gottfried Schatz Research Center, Molecular Biology and Biochemistry, Medical University of Graz, Neue Stiftingtalstraße 6/6, 8010 Graz, Austria.
[2] Gottfried Schatz Research Center, Cell Biology, Histology and Embryology, Medical University of Graz, Neue Stiftingtalstraße 6/2, 8010 Graz, Austria.
[3] BioTechMed Graz, Mozartgasse 12/2, Graz 8010, Austria. Correspondence and requests for materials should be addressed to W.F.G. (email: wolfgang.graier@medunigraz.at)

Biochemical signals in and out of mitochondria have to pass the outer (OMM) and inner (IMM) mitochondrial membrane. In this context, $Ca^{2+}$ transport has been particularly studied due to its importance for metabolic activity or initiating cell death programs[1]. While the OMM is quite permeable to ions, the movement of $Ca^{2+}$ across the IMM is strictly regulated by the Mitochondrial Calcium Uniporter Complex (MCU-Complex)[2,3]. This MCU-Complex consists of two pore-forming proteins: MCU (mitochondrial $Ca^{2+}$ uniporter)[4,5] and EMRE (essential MCU regulator)[6], along with the dominant-negative pore-forming subunit MCUb[7] and a scaffolding factor (MCUR1; mitochondrial calcium uniporter regulator 1)[8,9]. The activity of the MCU-Complex is controlled by MICU1 (mitochondrial calcium uptake 1)[10] and its paralog MICU2[11]. These regulatory proteins sense $Ca^{2+}$ via EF-hands located in the intermembrane space (IMS)[11,12]. The IMM is subdivided by cristae junctions (CJ) into two structurally, and functionally distinct domains, the cristae membrane (CM) and the inner boundary membrane (IBM)[13].

Distinct localizations of proteins between the CM and IBM were demonstrated for ATPase and electron transport chain subunits[14,15], and TIM proteins[13]. These reports emphasize a functional separation between the respiration machinery in the CM from other processes in the IBM. The CJ is established by the mitochondrial contact site and cristae organizing system (MICOS) and optic atrophy 1 (OPA1)[16,17]. As these proteins do not contain a $Ca^{2+}$ binding site, one might speculate that $Ca^{2+}$ elevations in the IMS do not affect the separation of proteins between the CM and IBM at the CJ. However, it is known that excessive $Ca^{2+}$ yields cytochrome c release from the CM[18], a phenomenon that is so far believed to depend on the formation of the permeability transition pore[19]. However, the exact mechanisms and effectors of $Ca^{2+}$-triggered cytochrome c release and whether or not, and if so, how an opening of the CJ is involved in this mechanism remains elusive.

The activation of the MCU-Complex is initiated by $Ca^{2+}$ binding to the EF-hand motifs of MICU1 yielding rearrangement of MICU1[12,20]. Although the molecular[20], dynamic[12] and post-translational[21] features of $Ca^{2+}$-induced MICU1 rearrangement are known, spatial distributions of MCU-Complex proteins are unknown. Moreover, whether the MCU-Complexes are already pre-assembled under resting conditions, or, whether assembly of the MCU-Complex requires $Ca^{2+}$-triggered binding to MICU1, is unclear.

Here, we exploit super-resolution structured illumination microscopy (SIM), electron microscopy and newly designed sub-mitochondrial $Ca^{2+}$ recordings to identify sub-mitochondrial distribution, dynamics, and localization-derived functions of MICU1 and MCU-Complex components under resting conditions and upon $Ca^{2+}$ mobilization. Our findings reveal a function of MICU1 as $Ca^{2+}$-dependent regulator of the CJ and of inner-cristae localization of cytochrome c. Altogether, our data present MICU1 as multi-functional regulator of an on-demand assembly of the MCU-Complex and mediator of $Ca^{2+}$ signaling to the cristae.

## Results

**MICU1 is exclusively localized in the IBM**. Applying super-resolution SIM, we first determined the sub-mitochondrial localization of MICU1. HeLa cells expressing MICU1 tagged with yellow fluorescent protein (YFP) were labeled with Mitotracker Red FM (MTR). Super-resolution images under resting conditions and after stimulation with histamine were acquired. These experiments show that MICU1 exclusively localizes to the IBM[15], and that this distribution does not change upon histamine stimulation (Fig. 1a, b). Morphological analysis of mitochondria showed no effect of MICU1-YFP expression (Supplementary

Fig. 1). To further establish that MICU1 is confined to the IBM, we co-transfected cells with MICU1-YFP and mCherry-TOM22. By analyzing the cross-section intensity profiles of the two fluorophores based on a fit of two superimposed Gaussian functions, we estimated a distance of $21 \pm 13$ nm (mean $\pm$ SD, $n = 105$) between the MICU1 and TOM22 (Fig. 1c–e; Supplementary Fig. 2 and 3), which is consistent with electron microscopy measurements[22]. Hence, by co-expression of MICU1-mCherry and MICU1-YFP a difference of $-5 \pm 11$ nm (mean $\pm$ SD, $n = 80$) was found. To validate these results, we measured diffraction-limited 100-nm-sized Tetraspec beads to analyze $x/y$ and $z$ chromatic abbreviation of our setup. For $x/y$, we determined a relative mean local chromatic abbreviation of 1.05% (Supplementary Fig. 4) while for $z$-axes a 45-nm shift was determined (Supplementary Fig. 5). The $z$-shift of 45 nm as well as dispersion measured with MTR- and MTG-labeled mitochondria in live cells only have a very minor effect of ~1.6 nm with regard to the $x/y$ measurements (Supplementary Figs. 6 and 7). Other proteins of the MCU-Complex (i.e., MCU, EMRE and UCP2) co-localized with Mitotracker Green (MTG) as IMM marker and were clearly separated from TOM22 (Supplementary Fig. 8).

**Design of various MICU1 mutants**. To evaluate the molecular basis of the IBM localization of MICU1, we designed MICU1 mutants that were either truncated to lack the poly-basic domain[10,20] (MICU1$^{1-70}$), both EF-hand motifs (MICU1$^{1-140}$), or the C-terminal helical oligomerization domain ($\Delta$C-MICU1) (Fig. 2a). In addition both EF hands were disabled by respective mutations (MICU1-$\Delta$EF)[12]. We also used MICU1-R455F-YFP and MICU1-R455K-YFP mutants, that either mimic or prevent methylation of MICU1[21] (Fig. 2a).

**MICU1 localization requires mitochondrial membrane potential**. The CJ enables the generation of high proton gradients across the CM[23,24]. By its poly-basic domain[10,20] MICU1 associates potentially with negatively-charged phospholipids (e.g. phosphatidylserine, cardiolipin[25]). Therefore, we examined the importance of a high proton gradient for MICU1 distribution by gradually reducing the mitochondrial membrane (IMM) potential ($\psi_{mito}$) with oligomycin A and antimycin A. For analyzing the sub-mitochondrial localization of MICU1-YFP a new quantification method based on a robust spatial thresholding approach was used (IBM association index) (Supplementary Fig. 9). The higher the IBM association index, the more the respective protein is localized to the IBM. This approach revealed that during depolarization MICU1 but not EMRE rapidly redistributed from the IBM into the entire IMM (Fig. 2b, c). Measurements of $\psi_{mito}$ using the potentiometric dye tetramethylrhodamine methyl ester perchlorate (TMRM) revealed that even partial depolarization caused diffusion of MICU1 into the CM (Fig. 2d, e). Because no correlated structural changes in mitochondrial morphology were observed (Supplementary Fig. 10), and MICU1 is not undergoing proteolytic cleavage even after 10 min of oligomycin A and antimycin A treatment (Supplementary Fig. 11), it is likely that the exclusive localization of MICU1 to the IBM is controlled by the $\psi_{mito}$.

**IBM localization of MICU1 is dependent on its poly-basic domain**. To examine the importance of the poly-basic domain of MICU1 for its IBM localization, the MICU1$^{1-140}$ and MICU1$^{1-70}$ mutants tagged to YFP (Fig. 2a) were transiently overexpressed in HeLa cells that were subsequently labeled with MTR. While MICU1$^{1-140}$ mimicked the IBM localization of wild-type MICU1 (Fig. 2f), MICU1$^{1-70}$ was located in the entire IMM (Fig. 2f). For

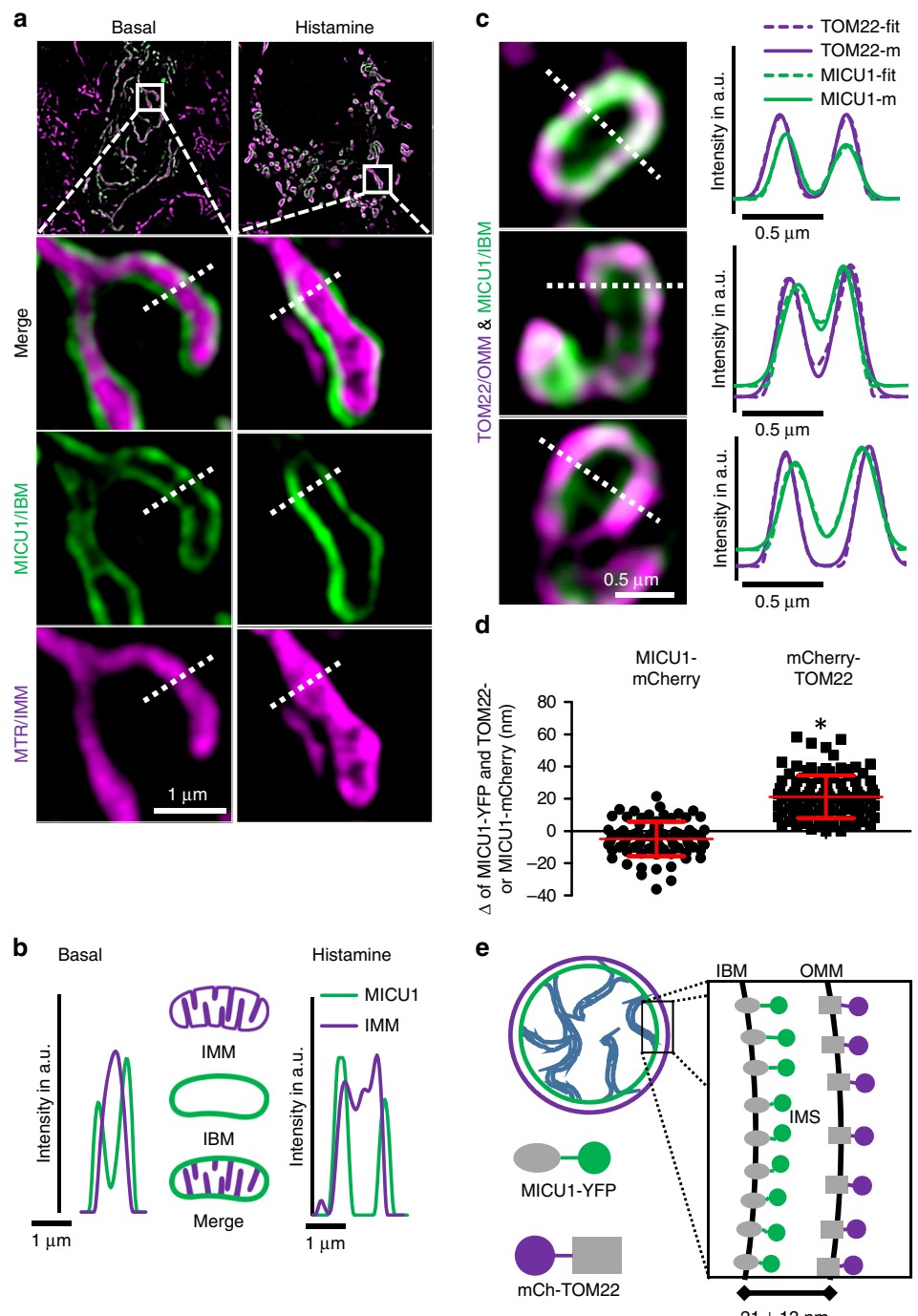

**Fig. 1** Super-resolution SIM microscopy localizes MICU1 to the IBM. **a** Cells were transiently transfected with MICU1-YFP (green), then stained with Mitotracker Red FM (MTR) (magenta) and examined using simultaneous dual-color 3D-SIM either under resting conditions, or 4 min after stimulation with 100 μM histamine. The upper panels provide an overall view of the mitochondria, and the dashed squares indicate the regions shown magnified below. The figures show merges of MICU1-YFP and MTR, along with MICU1-YFP (*MICU1*) and IMM (*MTR*) signals alone. No changes in the distribution of MICU1-YFP (nor IMM staining) were observed upon stimulation with histamine. **b** Line plots of MICU1 (green line) and the IMM staining with MTR (magenta line) of regions marked with dashed white lines in **a** are shown. A clear sub mitochondrial localization of MICU1 to the IBM is visible. **c** HeLa cells transfected with MICU1-YFP (green) and mCherry-TOM22 (magenta) were imaged using simultaneous dual-color 3D-SIM. Left panels show representative images. Right panels show line plots through mitochondria at locations indicated by the white dashed lines. By subjecting both channels (TOM22-m (magenta line), MICU1-m (green line)) to a double Gaussian-fit for each individual line (TOM22-fit (dotted magenta line), MICU1-fit (dotted green line) (right panels), the peaks of both distributions could be determined with sub-pixel accuracy. **d** HeLa cells transfected with MICU1-YFP and mCherry-TOM22 or MICU1-mCherry were analyzed regarding the relative distance between the YFP and mCherry distributions like shown in **c**. Data are shown as dot plots with the mean ± SD as red middle line and whiskers, respectively ($n_{mCherry-TOM22} = 105$, $n_{MICU1-mCherry}) = 80$). *$P < 0.05$ vs. respective control conditions carried out with unpaired double-sided *T*-test. **e** Schematic illustration of the mitochondrial inner boundary membrane and outer membrane showing the determined spacing between MICU1 and TOM22 of 21 ± 3 nm (mean ± SD). This value matches published distances between the OMM and IMM (lower right). Images and analyses were obtained from 2 to 5 mitochondria in at least 3 cells in each of 8 independent experiments ($n_{mCherry-TOM22} = 8/24/105$, $n_{MICU1-mCherry}) = 8/40/80$). Source data are provided as a Source Data file

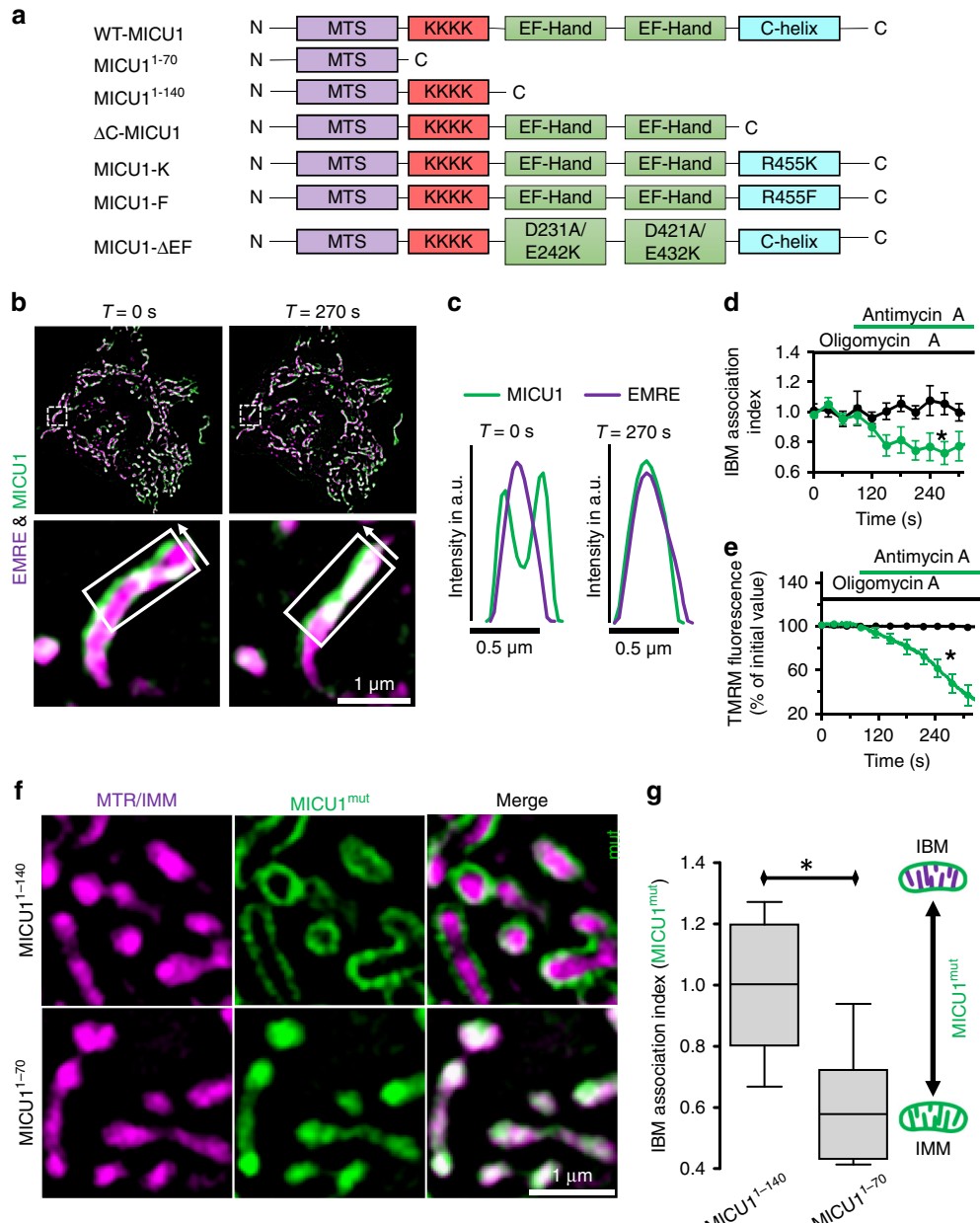

**Fig. 2** $\psi_{mito}$ and the poly-lysine domain of MICU1 are required for IBM localization. **a** Schematic representation of MICU1 mutants used in this work. (MTS→mitochondrial targeting sequence; KKKK→poly-lysine domain; EF-Hand→Ca$^{2+}$ binding domain; C-Helix→C-terminal hexamerization domain). **b** Representative images of SIM time-lapse experiment using HeLa cells transiently transfected with MICU1-YFP (green) and EMRE-mCherry (magenta) after $T = 0$ and $T = 270$ s. Cells were pre-incubated with oligomycin A (2 µM) and antimycin A (4 µM) was added at $T = 90$ s. **c** Line plots of MICU1 (green line) and EMRE (magenta line) of regions in between the white arrows in **b** confirm the redistribution of MICU1 into the IMM after addition of antimycin A. Differences between oligomycin A and oligomycin A plus antimycin A traces were measured from $T = 50$ s. **d** Time course of the MICU1 IBM association index over time after pre-incubation with oligomycin A with (green) or without (black) exposure to antimycin A starting at $T = 90$ s ($n_{Control} = 7$; $n_{Oligomycin/Antimycin} = 9$). Data are shown as the mean ± SEM. **e** Corresponding tetramethylrhodamine methyl ester perchlorate (TMRM) fluorescence, which indicates $\psi_{mito}$. Cells were pre-incubated with oligomycin A and exposed to antimycin A (green) or DMSO (black) at $T = 90$ s ($n_{Antimycin} = 8$; $n_{Antimycin/Oligomycin} = 9$). A significant drop in $\psi_{mito}$ after addition of antimycin A correlates with the drop in MICU1 IBM association index. Data are shown as the mean ± SEM. **f** Representative dual-SIM images of HeLa cells stained with MTR (magenta) and expressing MICU1$^{1–140}$-cpEGFP (green, upper panel) or MICU1$^{1–70}$-cpEGFP (green, lower panel). **g** Respective statistical analyses of the IBM association index of MICU1$^{1–140}$ and MICU1$^{1–70}$. The sub-mitochondrial distribution of MICU1$^{1–140}$ and MICU1$^{1–70}$ in the IBM and the CM was determined with the IBM association index. Horizontal lines represent the median, the lower and upper hinge show respectively first quartile and third quartile, and lower and upper whisker encompass minimal and maximal values. Images and analyses were obtained from $n = 9$ independent experiments, which assessed a total of 45 cells. *$P < 0.05$ vs. respective control conditions carried out with unpaired double-sided $T$-test. Source data are provided as a Source Data file

quantitative and statistical analyses the IBM association index was calculated (Fig. 2g). This different sub-mitochondrial localizations of both MICU1 mutants point to an important role of the protein's poly-lysine domain for its spatial location to the IBM. Recent reports shown that MICU1 is closely associated with cardiolipin[26]. Knockdown of taffazin (TAZ), an enzyme responsible for cardiolipin maturation[27], led to a rearrangement of MICU1-YFP into the entire IMM and slightly reduced mitochondrial form factor (Supplementary Fig. 12). These results show the correlation between the abundance of cardiolipin and the spatial distribution of MICU1.

**EF-hands and the methylation site do not contribute to MICU1 localization.** Next, we evaluated the role of the two EF-hand motifs of MICU1 on its IBM localization. Sub-mitochondrial localization of MICU1-ΔEF (Fig. 2a) was measured and the IBM association index was calculated. Disabling both EF-hands of MICU1 did only slightly reduce the proteins localization in the IBM (Supplementary Fig. 13). Because the apparent $Ca^{2+}$ binding affinity of MICU1 in response to R455 methylation by protein arginine methyltransferase 1 (PRMT1) is strongly attenuated[21], we examined whether R455 methylation may affect the IBM localization of MICU1. Therefore the distributions of the two respective mutations MICU1-R455F-YFP and MICU1-R455K-YFP[21] (Fig. 2a) were assessed and the IBM association index was calculated (Supplementary Fig. 13). Inhibition of PRMT1-mediated methylation (MICU1-R455K-YFP) and mimicking its stable methylation (MICU1-R455F-YFP) did not affect the proteins localization in the IBM (Supplementary Fig. 13). The general mitochondrial morphology regarding form factor, aspect ratio and size was also not changed by MICU1-R455F-YFP, MICU1-R455K-YFP or MICU1-ΔEF-YFP (Supplementary Fig. 14).

**MICU1 di-/multi-merization is necessary for IBM localization.** We next examined whether the quaternary structure of MICU1[12] contributes to its distribution. The sub-mitochondrial localization of the oligomerization site-lacking ΔC-MICU1-YFP mutant[20] was assessed (Fig. 2a). Notably, in cells with constitutive wild-type MICU1 ΔC-MICU1 was confined to the IBM (Fig. 3a). Depletion of endogenous MICU1 (Supplementary Fig. 15) yielded change of the distribution of the ΔC-mutant to the entire IMM (Fig. 3a). Mitochondrial morphology was not affected by ΔC-MICU1-YFP expression or knockdown of endogenous MICU1 (Supplementary Fig. 16).

**An intact cristae junction is essential for the IBM location of ΔC-MICU1.** The CJ is stabilized by several proteins, including OPA1, that forms pore-like structures[16,17]. To test whether the stability of the CJ is another important molecular determinant for MICU1 localization, OPA1 expression was reduced by using respective siRNA. Depletion of OPA1 that induced slight mitochondrial swelling (Fig. 3b, Supplementary Fig. 16) did not influence the distribution of wild-type MICU1 but allowed the ΔC-MICU1 mutant to localize in the CM even in the presence of endogenous MICU1 (Fig. 3b). Hence, the contribution of OPA1 on the CJ integrity was verified with transmission electron microscopy (TEM) that revealed a widening of the CJ by OPA1 knockdown (Fig. 4a, b, d). These data indicate that the exclusive IBM location of the ΔC-mutant of MICU1 requires a stable CJ.

**MICU1 contributes to the structural integrity of the CJ.** Our findings, showing that wild-type MICU1 did not enter the CM despite the knockdown of OPA1 and the dependency of the location of ΔC-MICU1 on the presence of wild-type MICU1 gave

us reason to believe that MICU1 may be involved in CJ stabilization. This was investigated with TEM where the effect of MICU1 depletion on the width of the CJ was obtained. Remarkably, similar to the knockdown of OPA1, silencing of MICU1 yielded strong widening of the CJ (Fig. 4a–d) (Control: $17.1 \pm 0.8$ nm; OPA1 knockdown: $26.9 \pm 1.3$ nm; MICU1 knockdown: $27.9 \pm 1.3$ nm; mean ± SEM, $n_{Control\ si} = 117$; $n_{OPA1\ si} = 97$; $n_{MICU1\ si} = 106$). Cristae widening by knockdown of OPA1 or MICU1 was largest at the interface to the IBM and got gradually smaller, while beyond 60 nm from the CJ no significant effect was found (Fig. 4d). To test for a contribution of mitochondrial $Ca^{2+}$ signaling to the CJ stabilization by MICU1, MCU and EMRE alone or together with MICU1 were silenced. MCU/EMRE knockdown did not change the CJ morphology and silencing of MCU/EMRE and MICU1 changed CJ morphology similar to the knockdown of MICU1 alone (Supplementary Fig. 17).

**CJ function of MICU1 for mitochondrial functional morphology.** We recently described a correlation between the CM movement and the mitochondria-ER proximity, the presence of OPA1 but not matrix $Ca^{2+}$ levels[28]. Accordingly, we analyzed the impact of silencing MICU1 on CM kinetics. Similar to the depletion of OPA1[28], knockdown of MICU1 yielded significant reduction of CM kinetics in the entire mitochondria (Supplementary Fig. 18).

Morphological analyses of mitochondria showed a reduction in mitochondrial branching and mitochondrial size upon depletion of MICU1. Using co-localization of MTG stained mitochondria and an ER-targeted RFP construct, we found that the number of mitochondria associated membranes (MAMs) remained unchanged (Supplementary Fig. 19). Cytochrome c release is a hallmark of apoptosis induction[29,30]. Considering the large widening of the CJ upon knockdown of either MICU1 or OPA1, the role of MICU1/OPA1 for cytochrome c localization was evaluated. Therefore, HeLa cells were depleted from MICU1 or OPA1, stained with Mitotracker Red CMXROS (MTR-CMX), fixed with paraformaldehyde (PFA) and immunostained for cytochrome c. Knockdown of either MICU1 or OPA1 yielded translocation of cytochrome c from the cristae into the IMS (Fig. 4e–g). Notably, cell viability and caspase-3/7 activity remained unchanged by knockdown of MICU1 or OPA1 (Supplementary Fig. 20).

**Mitochondrial ΔΨ and cristae $Ca^{2+}$ are protected by MICU1.** To estimate the physiological consequences of CJ widening by knockdown of OPA1 or MICU1 we measured mitochondrial cristae $Ca^{2+}$ and $\psi_{mito}$. Depletion of MICU1 or OPA1 but not MCU or EMRE reduced the $\psi_{mito}$ (Fig. 5a, Supplementary Fig. 21). Depolarization by knockdown of OPA1 or MICU1 was recovered by an overexpression of MICU1-YFP. In contrast to its wild-type form, expression of ΔC-MICU1-YFP was not able to re-establish $\psi_{mito}$ in MICU1/OPA1-depleted cells (Fig. 5a). These data suggest that silencing of OPA1 or MICU1 leads to structural loss in CJ integrity resulting in a leakage of $H_3O^+$ from cristae lumen (CL) and, thus, a reduction of $\psi_{mito}$ (Fig. 5b).

MICU1 was reported to serve as a gatekeeper to prevent mitochondrial matrix $Ca^{2+}$ overload[31,32]. Accordingly, we evaluated whether the CJ alteration under OPA1/MICU1 knockdown have implications on cristae and IMS $Ca^{2+}$ levels. For this purpose we exploited recently developed $Ca^{2+}$ sensors referred to as CL-GEMGeCO1 and IMS-GEMGeCO1 (M.W.-W., B.G., C.T. M.-S., C.K., H.B., E.E., R.M., W.F.G., manuscript submitted) to measure the $Ca^{2+}$ concentrations in the CL and the IMS. MICU1 and OPA1 knockdown increased basal $Ca^{2+}$ levels in the CL (Fig. 5c, Supplementary Fig. 22) but not IMS (Fig. 5d,

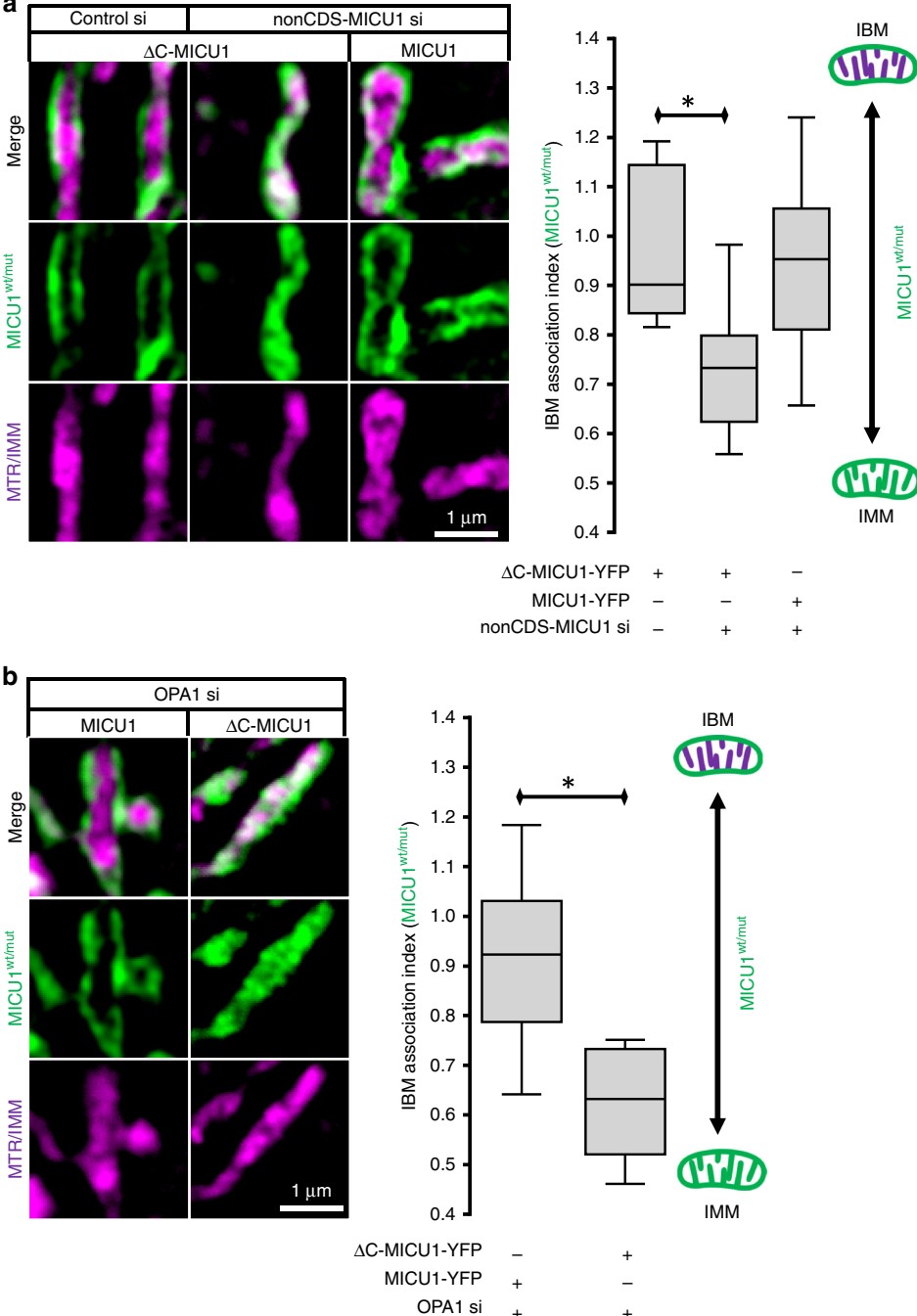

**Fig. 3** The C-terminal domain of MICU1 maintains IBM localization. **a** Left panel shows representative SIM images of HeLa cells stained with MTR (magenta) and expressing ΔC-MICU1-YFP (green) or MICU1-YFP (green) and transfected with either non-coding siRNA against MICU1 (nonCDS-MICU1 si) or control siRNA (Control si). In the right panel the respective statistical analyses of the distribution of MICU1 or ΔC-MICU1 after knockdown of constitutive MICU1 (nonCDS-MICU1 si) or OPA1 (OPA1 si) are shown. The sub-mitochondrial distribution of MICU1- and ΔC-MICU1-YFP in the IBM and the CM was determined in these experiments with the IBM association index ($n_{\Delta C-MICU1,control\ si} = 8$; $n_{\Delta C-MICU1,nonCDS\ MICU1\ si} = 10$; $n_{MICU1,\ nonCDS\ MICU1\ si} = 16$) . **b** Left panel shows representative SIM images of HeLa cells stained with MTR (magenta) and expressing MICU1-YFP (green) or ΔC-MICU1-YFP (green) and transfected with siRNA against OPA1 (OPA1 si). The IBM association index was determined for the distribution of MICU1-YFP and ΔC-MICU1-YFP with OPA1 knockdown in the right panel ($n_{MICU1,\ OPA1si} = 16$; $n_{\Delta C-MICU1,\ OPA1si} = 8$). Horizontal lines represent the median, the lower and upper hinge show respectively first quartile and third quartile, and lower and upper whisker encompass minimal and maximal values. *$P < 0.05$ vs. respective control conditions carried out with analysis of variance (ANOVA) with Bonferroni post hoc test or unpaired double-sided $T$-test. Source data are provided as a Source Data file

Supplementary Fig. 22). However, the Δ-ratio values after cell stimulation with histamine remained unchanged in the CL (Fig. 5c, Supplementary Fig. 22) and the IMS (Fig. 5d, Supplementary Fig. 22). In line with elevated basal cristae Ca²⁺,

basal mitochondrial matrix Ca²⁺ was significantly higher in OPA1- and MICU1-depleted cells (Supplementary Fig. 23). These results point to an increased CJ permeability for Ca²⁺ in cells lacking OPA1 or MICU1 (Fig. 5e).

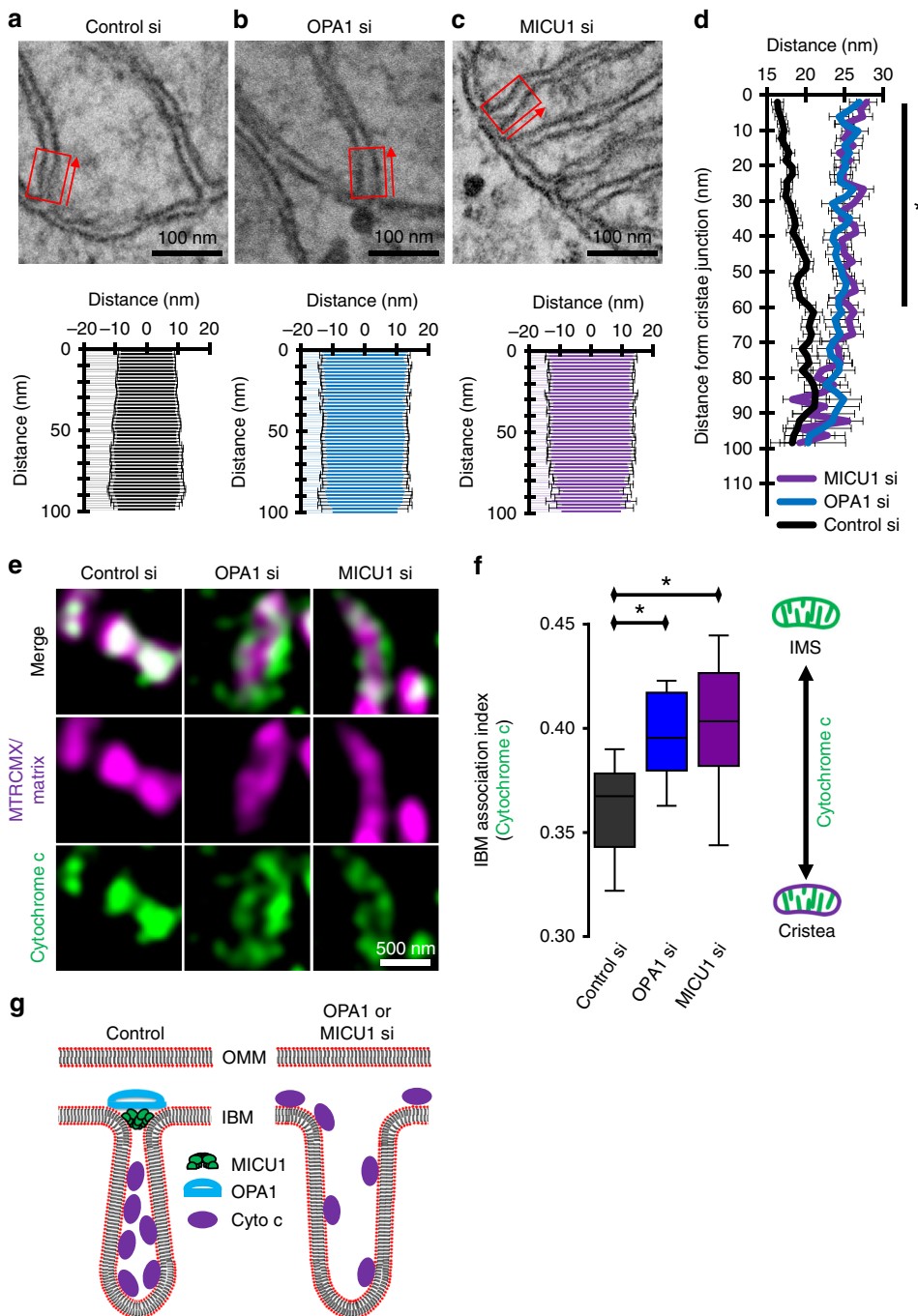

**Fig. 4** OPA1 and MICU1 tighten the CJ to restrict cytochrome c release from the cristae lumen. **a** The upper panel shows a single TEM image of a mitochondrion of HeLa cells transfected with control-siRNA (Control si). To analyze the topology of the cristae junction the cristae width was measured in 2 nm increments starting at the thought prolongation of the IBM and measured into the cristae as indicated with red boxes and arrows in **a**–**c**. In the lower panel the distance of the CM to the cristae center is plotted against the distance from the IBM ($n = 2/24/117$ with preparations/cells/CJ). **b** Same experiment as in **a** but with OPA1-siRNA (OPA1si) ($n = 2/22/97$ with preparations/cells/CJ). **c** Same experiment as in **a** but with MICU1-siRNA (MICU1si) ($n = 2/23/106$ with preparations/cells/CJ). **d** Comparative analysis of the cristae junction topology of Control (**a**, black), MICU1 (**c**, magenta) and OPA1 (**b**, blue) siRNA transfected cells. Data points represent the mean ± SEM ($n_{Control\ si} = 117$; $n_{OPA1\ si} = 97$; $n_{MICU1\ si} = 106$). **e** Representative images of HeLa cells transfected with Control, OPA1, or MICU1 siRNA, stained with MTRCMX, PFA-fixed and immunostained for cytochrome c. **f** Quantitative analysis of the sub-mitochondrial distribution of cytochrome c in HeLa cells transfected with Control, OPA1, or MICU1 siRNA ($n = 12$). Horizontal lines represent the median, the lower and upper hinge show respectively first quartile and third quartile, and lower and upper whisker encompass minimal and maximal values. **g** Schematic representation of the impact of the loss of MICU1 or OPA1 on mitochondrial cytochrome c redistribution caused by enlarged cristae junction. *$P < 0.05$ vs. respective control conditions carried out with analysis of variance (ANOVA) with Bonferroni post hoc test. Source data are provided as a Source Data file

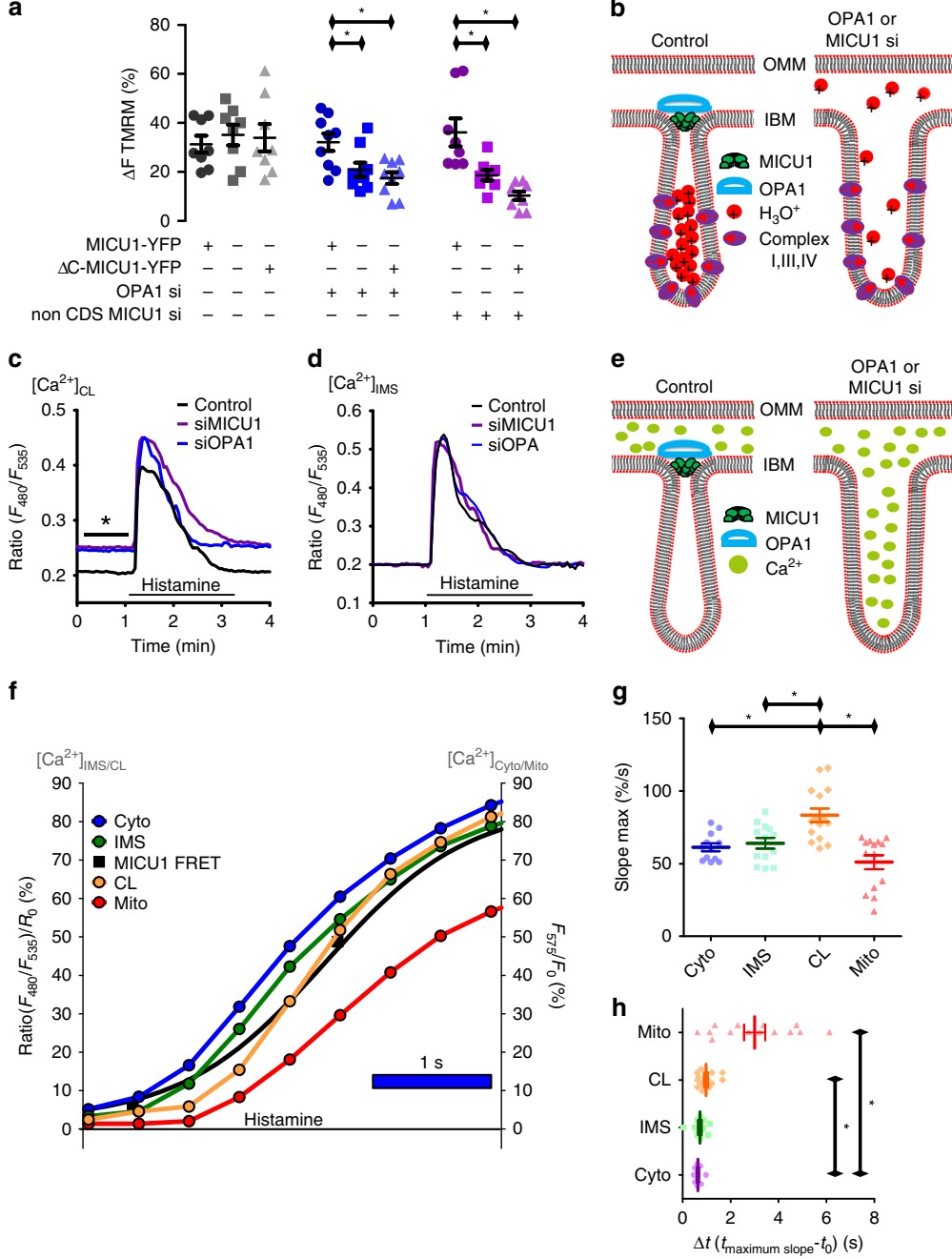

**Fig. 5** Knockdown of OPA1 and MICU1 decreases $\psi_{mito}$ and increase cristae $Ca^{2+}$. **a** HeLa cells expressing MICU1-YFP or ΔC-MICU1-YFP were transfected with Control (gray), OPA1 (blue), or MICU1 (magenta) siRNA. The TMRM Δfluorescence intensity after addition of 2 μM FCCP was measured ($n_{MICU1\text{-}YFP; \text{ Control si}} = 8$; $n_{no\text{-}FP; \text{ Control si}} = 8$; $n_{\Delta C\text{-}MICU1\text{-}YFP; \text{ Control si}} = 8$; $n_{MICU1\text{-}YFP; \text{ OPA1 si}} = 9$; $n_{no\text{-}FP; \text{ OPA1 si}} = 9$; $n_{\Delta C\text{-}MICU1\text{-}YFP; \text{ OPA1 si}} = 9$; $\underline{n}_{MICU1\text{-}YFP; \text{ nonCDS MICU1 si}} = 8$; $n_{no\text{-}FP; \text{ nonCDS MICU1 si}} = 8$; $n_{\Delta C\text{-}MICU1\text{-}YFP; \text{ nonCDS MICU1 si}} = 9$). **b** Schematic representation of the impact of MICU1 or OPA1 depletion on $\psi_{mito}$ caused by enlarged cristae junction. **c** HeLa cells were transfected with the cristae $Ca^{2+}$ sensor CL-GEMGeCO1 and Control (black), OPA1 (blue), or MICU1 (magenta) siRNA. Cells were challenged with 100 μM histamine to induce ER $Ca^{2+}$ release ($n_{Control si} = 9$; $n_{OPA1 si} = 10$; $n_{MICU1 si} = 10$). **d** HeLa cells were transfected with the IMS $Ca^{2+}$ sensor IMS-GEMGeCO1 and Control (black), OPA1 (blue), or MICU1 (magenta) siRNA. Cells were challenged with histamine ($n_{Control si} = 8$; $n_{OPA1 si} = 8$; $n_{MICU1 si} = 7$). **e** Schematic representation of the impact of loss of MICU1 or OPA1 on the sub-mitochondrial $Ca^{2+}$ redistribution reasoned by enlarged cristae junction. **f** HeLa cells were co-transfected with green IMS- or CL-GEMGeCO1 and red CARGeCO1 or mt-CARGeCO1 for a temporal correlation of $Ca^{2+}$ elevations upon stimulation with 100 μM histamine in the absence of extracellular $Ca^{2+}$. Average curves of dynamic $Ca^{2+}$ increases in the cytosol (blue curve), the IMS (green curve), the CL (orange curve) or the mitochondrial matrix (red curve) in comparison with a representative sigmoidal fitted MICU1-FRET curve obtained from published data (black)[12]. **g** Statistical evaluation of maximal slopes calculated in percentage per second (%/s) and **h** duration of lag time evaluated from onset of cytosolic $Ca^{2+}$ rise to maximal slope (Δt) between $[Ca^{2+}]_{Cyto}$ (Cyto, blue bar, $n = 12$), $[Ca^{2+}]_{IMS}$ (IMS, dark green bar, $n = 13$), $[Ca^{2+}]_{CL}$ (CL, orange bar, $n = 15$) and $[Ca^{2+}]_{Mito}$ (Mito, red bar, $n = 14$). Data shown are dot plots with the mean ± SEM as middle line and whiskers. *$P < 0.05$ vs. respective control conditions carried out with analysis of variance (ANOVA) with Bonferroni post hoc test. Source data are provided as a Source Data file

**Spatiotemporal correlation of sub-mitochondrial Ca$^{2+}$ entry.** The kinetics of sub-mitochondrial Ca$^{2+}$ signaling upon histamine were analyzed using the respective Ca$^{2+}$ sensor (Supplementary Fig. 24). As shown in Fig. 5f, cytosolic Ca$^{2+}$ elevation was instantly followed by Ca$^{2+}$ elevations in the IMS. The Ca$^{2+}$ signal in the CL developed slowly but reached maximum in the same time than the cytosol and the IMS (Fig. 5g). This delayed onset of CL Ca$^{2+}$ elevation correlated with the rearrangement of MICU1 heteromers (MICU1-FRET data are reused from a previous publication[12]). Elevation of matrix Ca$^{2+}$ was delayed compared with all other regions and showed the highest diversity within cells (Fig. 5f, h, Supplementary Fig. 24).

**Upon Ca$^{2+}$ mobilization MICU1 anchors MCU/EMRE at the IBM.** The discovery of MICU1[10] was followed by the identification of MCU[5,6] that co-immunoprecipitate with each other[5]. We next determined the distribution of MCU within the IMM. Using MCU-mCherry together with MICU1-YFP we found that, in contrast to MICU1, MCU is uniformly distributed throughout the entire IMM in resting cells (Fig. 6a, b). However, after stimulation with histamine, MCU joins MICU1 to predominantly localize in the IBM. Ca$^{2+}$-induced redistributions of MCU were not observed in cells exclusively expressing the ΔC-mutant of MICU1 that does not bind MCU[20] (Fig. 6a, b, Supplementary Fig. 25). Excitingly, mitochondrial matrix Ca$^{2+}$ elevation upon histamine strictly correlates with the shift in the distribution of MCU to the IBM (Fig. 6c, d). Our results additionally revealed that the redistribution of MCU is independent of extracellular Ca$^{2+}$ (Supplementary Fig. 26) and requires MICU1 (Supplementary Fig. 27). We further performed experiments to validate the MCU-shuttling process and mitochondrial morphology in HEK, A549 and MCF-7 cells (Supplementary Fig. 28). All cell types showed the same phenomena of Ca$^{2+}$-induced MCU shuttling to MICU1 in the IBM. Notably, upon histamine, HeLa and HEK cells undergo mitochondrial fission, MCF-7 cells show slight swelling but no fragmentation, and A549 cells did not change their morphology (Supplementary Fig. 29). These findings indicate that cytosolic Ca$^{2+}$ triggers anchoring of MCU to MICU1 in the IBM in non-excitable cell lines.

Next, we mapped the distribution of the second pore-forming protein of the MCU-Complex, EMRE[6]. An EMRE-mCherry construct was co-expressed with MICU1-YFP. Under basal conditions, EMRE was distributed throughout the entire IMM and this distribution did not change upon stimulation with histamine (Fig. 7a, b). Since EMRE-MCU stoichiometry could contribute to this result, we pursued experiments involving the co-expression of MCU, MICU1-YFP, and EMRE-mCherry. Under these conditions, EMRE accumulated in the IBM upon Ca$^{2+}$ mobilization (Fig. 7c, d). The physiological functionality of EMRE-mCherry was validated by recovery experiments in EMRE-KO cells (Supplementary Fig. 30). Furthermore, co-immunoprecipitation of EMRE-mCherry-HIS and MCU-mCherry-HIS with MICU1-FLAG showed direct interaction of these constructs (Supplementary Fig. 30). Notably, MICU1-YFP expression did not change the arrangement of the IMM-marker MTR[33], indicating that the integrity of the IMM was maintained throughout our experimental conditions (Supplementary Fig. 31).

**UCP2 facilitates MCU anchoring to methylated MICU1 in the IBM.** Uncoupling protein 2 (UCP2) specifically binds to PRMT1-methylated MICU1[21], normalizes the Ca$^{2+}$ sensitivity of MICU1 and restores mitochondrial Ca$^{2+}$ influx[21]. Accordingly, we examined the roles of PRMT1 and UCP2 in sub-mitochondrial MCU distribution during Ca$^{2+}$ mobilization. In UCP2 knockout (UCP2-KO) cells transiently expressing MICU1 MCU-mCherry

did not localize to the IBM in response to histamine (Fig. 8a). Furthermore, mitochondrial Ca$^{2+}$ uptake upon histamine was strongly reduced in UCP2-KO cells (Supplementary Fig. 31), despite strong cytosolic Ca$^{2+}$ signals (Supplementary Fig. 32). Knockdown of PRMT1 in UCP2-KO cells, however, restored Ca$^{2+}$-evoked MCU accumulation in the IBM (Fig. 8a). Over-expression of PRMT1 in control or UCP2-KO cells did not affect sub-mitochondrial localization of MCU or MICU1 (Supplementary Fig. 33).

Our next experiments examined the distribution of UCP2-mCherry in Mitotracker Green (MTG)-stained cells. For this purpose we used HeLa cells that have high levels of methylated MICU1 and therefore require UCP2 for MCU-Complex activation[21]. Under basal conditions UCP2 localizes in the entire IMM and accumulates in the IBM upon histamine (Fig. 8b, c). Depletion of either MICU1 or PRMT1 prevented this Ca$^{2+}$-evoked redistribution of UCP2 (Fig. 8b, c), pointing to a dynamic interaction of methylated MICU1 with UCP2. Over-expression of PRMT1 did not change the shuttling characteristics of UCP2 or MCU in HeLa or UCP2-KO cells (Supplementary Figs. 33 and 34). In agreement with our previous work[21], depletion of PRMT1 in cells expressing the methylation mimicking MICU1-R455F-YFP mutant did not affect the accumulation of UCP2 upon histamine, while an expression of the methylation-resistant MICU1-R455K-YFP did not show any UCP2 translocation to the IBM (Supplementary Fig. 35). Neither UCP2-mCherry nor PRMT1 knockdown affected mitochondrial morphology in WT-MICU1-YFP, MICU1-R455F-YFP or MICU1-R455K-YFP expressing cells (Supplementary Fig. 36).

## Discussion
Considering the Ca$^{2+}$-induced rearrangement of MICU1[20] as an essential step in the activation of the MCU-Complex[12], the concept that the MCU-Complex consists as a rather stable complex awaiting its activation by elevation of Ca$^{2+}$ in the IMS is challenged[9]. Notably, our recent work, introducing super-resolution SIM on sub-organelle membrane dynamics, described a unique regulation of cristae dynamics by spatial Ca$^{2+}$ inside the MAMs[28]. In this work, we utilized live-cell SIM and combined it with transmission electron microscopy[34] and newly designed tools for spatial Ca$^{2+}$ measurements to investigate the spatial dynamics of the MCU-Complex and the organization of the IMM under resting conditions and upon cell stimulation.

Applying dual-color SIM, a surprising shell-like distribution of MICU1 but not of the other MCU-Complex components MCU, EMRE and UCP2 was found that markedly differed from the IMM structure. The localization of MICU1 was confirmed lately by Proteinase K and carbon extraction assays to be at the outer leaflet of the IMM facing the IMS[35]. However, the comparison with OMM marker TOM22 revealed a distance of 21 ± 13 nm (mean ± SD) between this protein and MICU1, which is consistent with reported measures using electron microscopy[22]. These data indicate that MICU1[20] is exclusively localized in non-cristae structure of the IMM, referred as IBM[13]. Because MICU1 remained in the IBM even upon intracellular Ca$^{2+}$ release that triggers rearrangement of MICU1 to dimers[12], it is tempting to speculate that the specific location of MICU1 in the IBM does not depend on the status of oligomerization but the protein's structure.

Because depolarization of the IMM by oligomycin A and antimycin A yielded a redistribution of MICU1 into the entire IMM, we speculate that MICU1 because of its positively charged poly-lysine domain may not enter the cristae which is positively charged by the accumulation of H$_3$O$^+$ inside the cristae due to the activity of the respiration chain[13–15]. Our findings that all MICU1 variants (i.e. MICU1$^{1–140}$, MICU1-F, MICU1-K, MICU1-

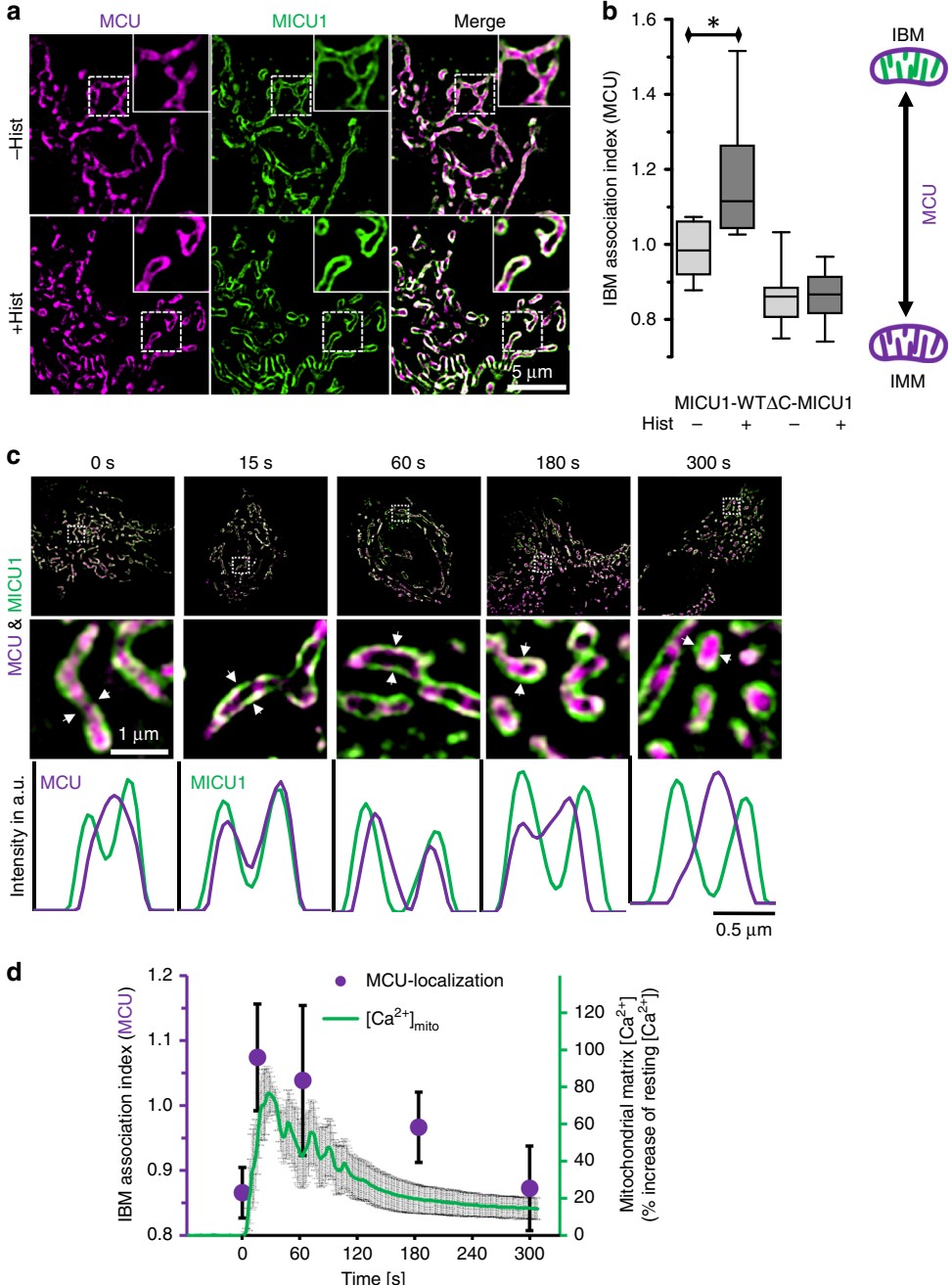

**Fig. 6** MCU redistributes with MICU1 correlative to intermembrane Ca$^{2+}$ spikes. **a** HeLa cells were transiently transfected with MCU-mCherry and MICU1-YFP and imaged with SIM under resting conditions (−Hist) and 90 s after stimulation with histamine (100 μM) (+Hist). Representative SIM captures showing the distribution of MCU-mCherry (magenta) and MICU1-YFP (green) are presented. Dotted squares indicate areas shown at high magnification in the upper right corner. **b** Statistical analysis of the sub-mitochondrial MCU-mCherry distribution in MICU1- or ΔC-MICU1-YFP- expressing cells represented as IBM association index ($n = 8$). Horizontal lines represent the median, the lower and upper hinge show respectively first quartile and third quartile, and lower and upper whisker encompass minimal and maximal values. **c** Representative images (top and middle) and line plots (bottom) of MCU-mCherry (magenta) and MICU1-YFP (green) under resting conditions (0 s) and 15, 60, 180 and 300 s after stimulation with histamine (100 μM) in nominally Ca$^{2+}$-free buffer. **d** Time correlation between the accumulation of MCU in the IBM (magenta points) and mitochondrial Ca$^{2+}$ concentration (green line and gray error bars) upon stimulation with 100 μM histamine ($n = 9$). Cells were transiently transfected with mito-R-GECO1 to measure mitochondrial matrix Ca$^{2+}$ concentrations upon histamine (green line) ($n = 8$). Data are shown as mean ± SEM as middle line and whiskers, respectively. *$P < 0.05$ vs. respective control conditions carried out with unpaired double-sided $T$-test. Source data are provided as a Source Data file

ΔEF, ΔC-MICU1) that contain the poly-lysine domain (Fig. 2a) are located to the IBM while the poly-lysine domain lacking MICU1$^{1–70}$ mutant was found throughout the entire IMM, indicate that sub-mitochondrial localization of MICU1 depends on the poly-lysine domain and its ability of electrostatic repulsive power. A potential interaction partner of MICU1 for spatial electrostatic interaction might be cardiolipin[25]. Notably, by its pKa of 7.5 cardiolipin may facilitate a localization-dependent dimorphism of charged head groups separated into the IBM and the CM with a pH of ~7.4 and <7.4, respectively[26,36]. The

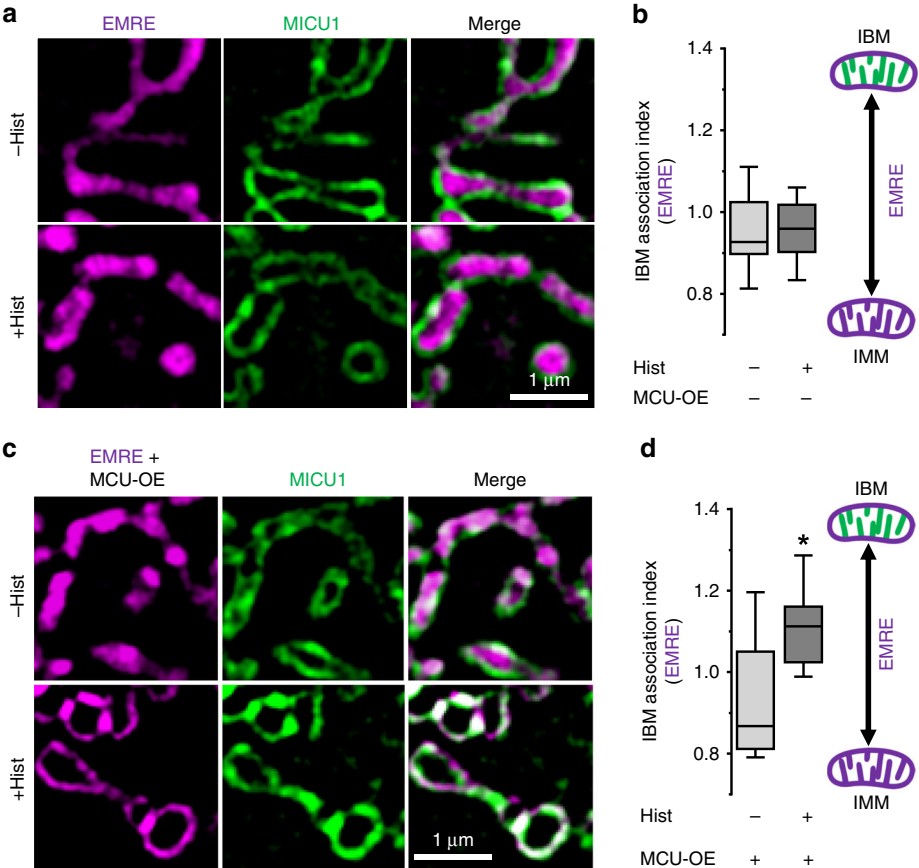

**Fig. 7** EMRE redistributes in a MCU-dependent manner to the IBM under ER Ca$^{2+}$ release. **a** Representative SIM images of HeLa cells that were transiently transfected with EMRE-mCherry (magenta) and MICU1-YFP (green) under resting conditions and 90 s after stimulation with 100 µM histamine. **b** The sub-mitochondrial localization of EMRE-mCherry under resting conditions (−Hist) and 90 s after treatment with histamine (100 µM) (+Hist) was determined by the IBM association index ($n = 8$). **c** Identical experiments as those in **a** but using cells that overexpressed wild-type MCU. **d** Identical experiments as those in **b** but using cells that overexpressed wild-type MCU ($n = 8$). Horizontal lines represent the median, the lower and upper hinge show respectively first quartile and third quartile, and lower and upper whisker encompass minimal and maximal values. Images and analyses were obtained from at least 10 cells in each of 8 experiments. *$P < 0.05$ vs. respective control conditions carried out with unpaired double-sided $T$-test. Source data are provided as a Source Data file

hypothesis of cardiolipin-dependent MICU1 restriction to the IBM is strengthened by our results showing MICU1 rearrangement into the entire IMM when cells were silenced for taffazzin[27]. The high sensitivity of MICU1 to depolarization might explain the ubiquitous IMM distribution of MICU1 in APEX2-based electron microscopy, in which depolarization during fixation might have influenced the sub-organellar localization of MICU1 (ref. [37]).

MICU1 exists either as dimer or as hexamer via its C-terminal oligomerization site[20]. We found that ΔC-MICU1, which does not form multimers[20], is restricted to the IBM in the presence of endogenous MICU1 but distributed through the entire IMM in cells depleted of endogenous MICU1. These data indicate that the oligomeric MICU1 affects the distribution of the ΔC-MICU1 mutant by stabilizing the CJs. The CJ is known to be established by several proteins, including OPA1/OMA1[16,17]. Our findings that knockdown of OPA1 did not change the exclusive IBM location of wild-type MICU1 but facilitated the redistribution of ΔC-MICU1 into the CM even in the presence of endogenous MICU1 suggest a model in which the sub-mitochondrial localization of MICU1 to the IBM is dependent on the stability of the CJ. In turn, MICU1 regulates the CJ stability pointing to an IBM self-containing function of MICU1. This assumption on a new feature of MICU1 is further supported by our electron microscopy data that revealed a MCU/EMRE-independent widening of

the CJ by ~58% under conditions of MICU1 knockdown, similar to the depletion of OPA1[38,39].

The contribution of MICU1 to CJ integrity is further supported by a series of functional assessments and respective reports in the literature: first, similar to our recent findings that widening of the CJ by knockdown of OPA1 yields reduction in CM-kinetic[28], MICU1 knockdown also reduced CM kinetics. Second, diminution of MICU1 resulted in higher mitochondrial fragmentation and lower branching, an effect that also occurs upon depletion of OPA1[28,40]. Third, our data reveal that similar to a knockdown of OPA1, depletion of MICU1 yields relocalization of cytochrome c from the cristae into the IMS (Fig. 4g). These findings are coherent with reports showing that knockdown of OPA1 and MICU1 enhance susceptibility for the induction of apoptosis[17,38,41,42]. Fourth, we demonstrate that knockdown of either OPA1 or MICU1 results in mitochondrial membrane depolarization. The known localization of ATPase and proteins of the electron transport chain within the CM[14,15] yields a proton accumulation within the CL[24]. Accordingly, IMM depolarization upon knockdown of OPA1 or MICU1 may be a consequence of the CJ widening and a subsequent dilution of the CM's proton gradient (Fig. 5b). Fifth, our findings that the depolarization upon MICU1/OPA1 knockdown is prevented by expression of MICU1-YFP, but not by the C-terminal lacking ΔC-MICU1-YFP mutant support the assumption of the contribution of MICU1 to CJ

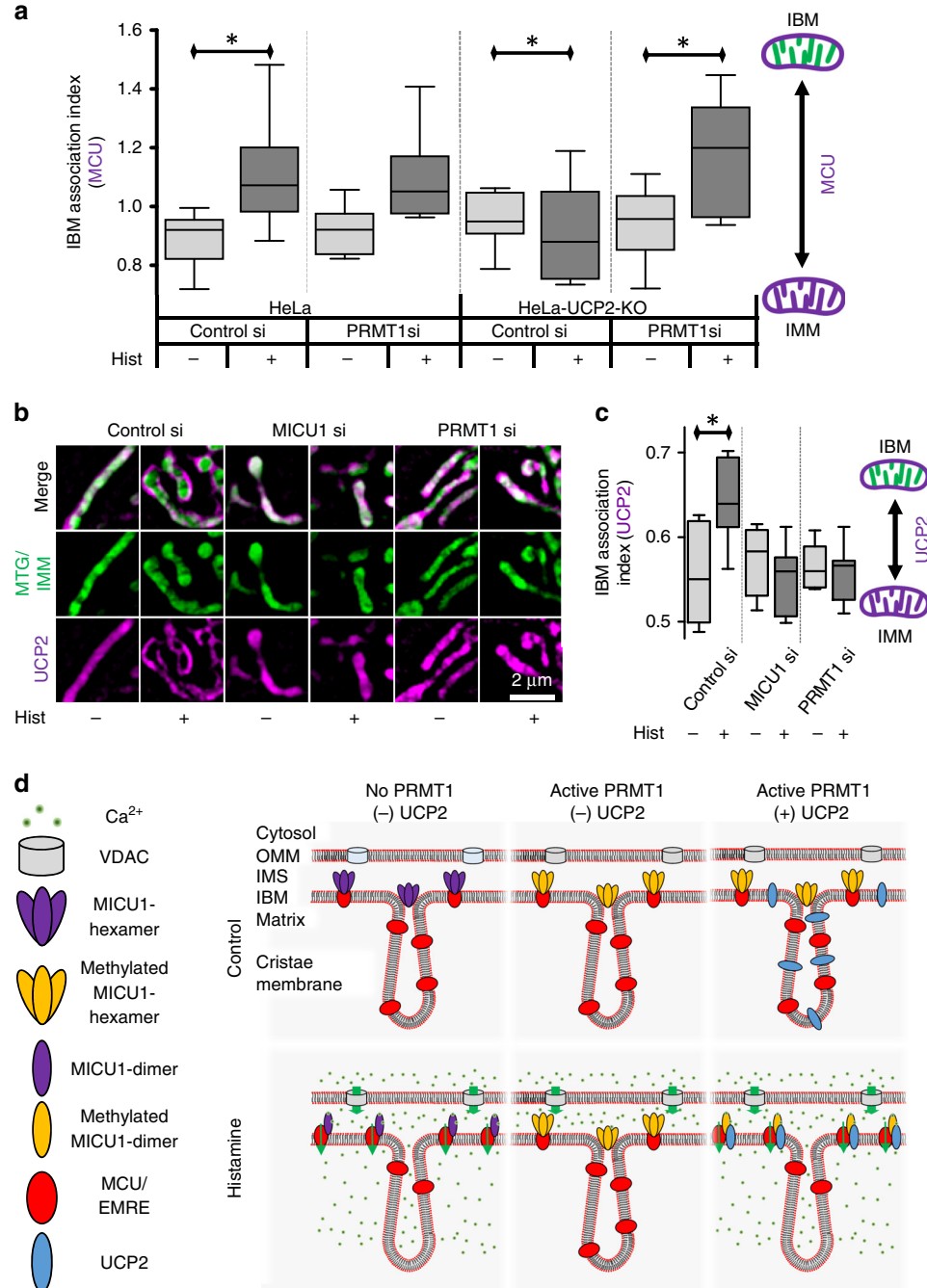

**Fig. 8** Under elevated PRMT1 activity, UCP2 is essential for MCU IBM relocation. **a** Statistical analyses of sub-mitochondrial localization of MCU-mCherry were performed on control (*HeLa*) and UCP2$^{-/-}$ (*HeLa-UCP2-KO*) cells with and without PRMT1 knockdown under resting conditions and 90 s after stimulation with 100 μM histamine (*n* = 8). **b** Representative SIM images of cells expressing UCP2-mCherry (magenta) and stained with MTG (green) in the presence (*Control si*) or absence of endogenous MICU1 (*MICU1 si*) or PRMT1 (*PRMT1 si*) under resting conditions (−) and 90 s after stimulation with 100 μM histamine (+). **c** Analysis of the sub-mitochondrial localization of UCP2 during the experiments described in **b** (*n* = 8). **d** Model for the sub-mitochondrial localization of MICU1, and localization and assembly of the MCU-Complex under resting conditions (*Control*) and upon IP$_3$-mediated intracellular Ca$^{2+}$ release (*Histamine*) in cells with low PRMT1 activity (left), in cells with active PRMT1 lacking UCP2 (middle), and in cells with active PRMT1 and in the presence of UCP2 (right). In the absence of PRMT1 activity, Ca$^{2+}$ spikes in the intermembrane space restructure MICU1 hexamers into dimers in the IBM, where they act as diffusion traps for MCU. These molecules can now anchor EMRE in the IBM, thus assembling MCU-Complexes there. In the absence of UCP2, PRMT1 prevents the reorganization of MICU1 into dimers upon intracellular Ca$^{2+}$ release, thus no shuttling of MCU (nor EMRE) to the IBM takes place and no functional MCU-Complexes are assembled. However, the presence of UCP2 allows the dimerization of MICU1 even in the presence of elevated PRMT1 activity, and thus restores the diffusion trap capabilities of MICU1 by either stabilizing the methylated MICU1 dimer or preventing the methylated MICU1 oligomer from disrupting it. Consequently, the MCU-Complex is assembled and mitochondrial Ca$^{2+}$ uptake is restored. Analyses were obtained from at least 5 cells each in 8 experiments. Horizontal lines represent the median, the lower and upper hinge show respectively first quartile and third quartile, and lower and upper whisker encompass minimal and maximal values. *$P$ < 0.05 vs. respective control conditions carried out with unpaired double-sided *T*-test. Source data are provided as a Source Data file

integrity. Sixth, MICU1 knockdown leads to increased oxygen consumption rate[43], possibly reflecting membrane depolarization. Seventh, by using newly developed $Ca^{2+}$ sensors (M.W.-W., B.G., C.T.M.-S., C.K., H.B., E.E., R.M., W.F.G., manuscript submitted) that exclusively measure $Ca^{2+}$ inside the cristae or the IBM we demonstrate that weakening of the CJ integrity by either OPA1 or MICU1 knockdown yields an increase in resting $[Ca^{2+}]$ levels in the cristae but not IMS. These findings are in line with reports showing that the initial $Ca^{2+}$ uptake velocity was increased in cells depleted by either OPA1 or MICU1[41,44]. Eighth, our findings that cristae $Ca^{2+}$ rapidly increased upon intracellular $Ca^{2+}$ release even in the presence of MICU1 indicate that MICU1 serves as $Ca^{2+}$ regulated gatekeeper for $Ca^{2+}$ in the CJ. Ninth, similar to knockdown of OPA1, diminution of MICU1 increases resting matrix $Ca^{2+}$, most likely because the CJ widening allows $Ca^{2+}$ to enter the cristae where active MCUs (i.e., not negatively controlled by MICU1) are located. Tenth, our temporal $Ca^{2+}$ propagation and MICU1 disassembly experiments highlight the role of MICU1 as a $Ca^{2+}$-dependent stabilizer of the CJ. Altogether, our data strongly point to a significant function of MICU1 by contributing to integrity of the CJ (Fig. 5e).

Our data demonstrate that, in contrast to the exclusive IBM distribution of MICU1, MCU and EMRE localize to the entire IMM. However, after stimulation with histamine MCU colocalized with MICU1 in the IBM, thus, indicating a dynamic assembly of the MCU-Complex upon cytosolic/IMS $Ca^{2+}$ elevation exclusively in the IBM. Our findings that EMRE-mCherry only relocalizes into the IBM upon histamine stimulation if MCU was simultaneously overexpressed, points to a strict stoichiometry for these proteins. Co-immunoprecipitation experiments show a direct interaction of MICU1 and EMRE, potentially by electrostatic interaction of the poly-basic domain of MICU1 with the C-terminus of EMRE[45,46]. That disagrees with our finding that EMRE is only shuttling to the IBM in a MICU1 and MCU-dependent manner. Vais et al. showed that the C-terminal tail of EMRE is oriented into the mitochondrial matrix, not able to interact with IMS localized MICU1[47]. Thus, we speculate that after the lysis process for protein extraction, integrity of membrane structures is affected, thus, misleadingly allow interactions of proteins that do not interact under physiological conditions due to their trans-membrane orientations. In case of EMRE/MICU1, this possibility is even more likely since both proteins are part of the MCU-Complex and are in close proximity. Furthermore, the poly-basic domain of MICU1 is potentially crucial for the interaction of MICU1 with IMM phospholipids (e.g., cardiolipin[25]) and its IBM localization.

The specificity of MCU/EMRE redistribution is indicated by our findings that MCU/EMRE do not relocalize in the exclusive presence of ΔC-MICU1 that does not bind MCU[20]. The dynamic and reversible assembly of MCU-Complex to harbor MCU by the MICU1 dimers in the IBM is further demonstrated by our results on the strict correlation of mitochondrial matrix $Ca^{2+}$ elevation with the IBM localization of MCU. Consistent with our findings, a recent report describes that the permanent activation of MCU by hydrogen peroxide also causes its accumulation in the IBM[48]. Our results also show that the redistribution of MCU/EMRE is independent of extracellular $Ca^{2+}$ but requires a rise in cytosolic/IMS $Ca^{2+}$ and the presence of MICU1. Our data demonstrate that IMS $Ca^{2+}$ increases trigger the dissociation of MICU1[12,20] that, subsequently serves as diffusion trap for MCU that, subsequently, anchors EMRE. While the shuttling process of MCU was confirmed by us in different non-excitable cells (HeLa, HEK, A549, or MCF-7 cells), excitable cells like neurons or cardiomyocytes might rely on different regulatory mechanisms for mitochondrial $Ca^{2+}$ uptake (e.g. various stoichiometries of the MCU-Complex members)[49], posttranslational modifications

(phosphorylation, methylation, and oxidation)[21,35,48], or the abundance of cardiolipin[3].

Our recent findings demonstrate that PRMT1-mediated asymmetric methylation of MICU1 decreases its $Ca^{2+}$ binding affinity yielding reduced mitochondrial $Ca^{2+}$ uptake[21]. However, UCP2 specifically binds to methylated MICU1, and, consequently, restores its $Ca^{2+}$ sensitivity and, thus, MCU-Complex activity[21]. In the present work using HeLa cells with high PRMT1 activity[21], no MCU shuttling was found in UCP2-KO HeLa cells after cell stimulation. However, knockdown of PRMT1 recovered MCU shuttling in UCP2-KO cells. These data confirm our previous results and further demonstrate that mitochondrial $Ca^{2+}$ uniporter function depends on UCP2 in cell types with elevated PRMT1 activity[21]. The sub-mitochondrial localization of UCP2 has not been previously explored. While UCP4 can localize to the IBM[24], our data revealed that UCP2 is in the entire IMM under resting conditions and translocates upon cell stimulation into the IBM in a MICU1- and PRMT1-dependent manner.

In accordance to our previous work[21], we examined the role of MICU1 methylation for UCP2 translocation using MICU1-R455F and MICU1-R455K mutants that either mimic or prevent methylation of R455, respectively[21]. Our findings that UCP2 translocation to the IBM upon ER-$Ca^{2+}$ release was independent of PRMT1 when MICU1-R455F was co-expressed but disabled in the presence of MICU1-R455K, indicate that methylation of MICU1 at R455 is prerequisite for the relocation of UCP2 to the IBM. They also confirm a fundamental role for UCP2 in the assembly and activation of the MCU-Complex under conditions of elevated PRMT1 activity[21,50].

Overall, our present data change the current model of organization, localization and processes of stimulation of the MCU-Complex (Fig. 8d). In particular, our findings demonstrate that MICU1 is exclusively localized in the IBM due to its hexamerization and poly-lysine domain. Hence, MICU1 exhibits essential function by contributing to CJ stability. By this newly discovered function of MICU1, the protein is involved in maintaining $\psi_{mito}$, protection against matrix $Ca^{2+}$ overload, and restriction of cytochrome c to cristae. Therefore, a new model of MCU-Complex activation is presented in which MICU1 dissociates into dimers serving as diffusion trap for MCU in the IBM. The IBM-harbored MCU subsequently anchors EMRE, thereby establishing functional MCU-Complexes exclusively at this location. PRMT1 activity and MICU1 methylation prevent MICU1 dimerization and MCU-Complex assembly, but UCP2 that interacts with methylated MICU1 facilitates MCU-Complex assembly in the IBM. This new model seamlessly meshes existing data with new findings and provides an advanced perspective on the dynamic organization of mitochondrial ion regulation and how MICU1 contributes to CJ integrity.

## Methods

**Single and dual camera SIM imaging.** The SIM setup used is composed of a 405-, 488-, 515-, 532- and a 561-nm excitation laser introduced at the back focal plane inside the SIM box with a multimodal optical fiber. For super-resolution, a CFI SR Apochromat TIRF ×100-oil (NA 1.49) objective was mounted on a Nikon-Structured Illumination Microscopy (N-SIM) System with standard wide field and SIM filter sets and equipped with two Andor iXon3 EMCCD cameras mounted to a Two Camera Imaging Adapter (Nikon Austria, Vienna, Austria). For calibration and reconstruction of SIM images the Nikon software Nis-Elements was used. Reconstruction was always done with the same robust setting to avoid artifact generation and ensures reproducibility with a small loss of resolution of 10% compared to most sensitive and rigorous reconstruction settings. Prior to each measurement, laser adjustment was checked by projecting the laser beam through the objective at the top cover of the bright field arm of the microscope. Lasers were aligned and focus using the interlock system screws at the N-SIM box to ensure appropriate illumination of the sample. 100 nm Tetraspec beads (Invitrogen, Thermo Fischer Scientific, Vienna, Austria) were diluted 1:200 in 1 ml 0.01% poly-L-lysine and incubated on the 1.5 H high-precision glass cover slips (Marienfeld-Superior, Lauda Königshofen, Germany) for 20 min. Afterwards, the plate was

washed once and transferred to the live-cell chamber containing 1 ml of buffer (2CaB) containing in mM: 2 $CaCl_2$, 140 NaCl, 5 KCl, 1 $MgCl_2$, 1 HEPES and 10 D-Glucose, pH adjusted to 7.4 (all buffer salts were obtained from Roth, Graz, Austria). Ring correction was done as follows: 3D stacks of beads were acquired to verify the point spread function of the system. Potential asymmetric pointspread function was corrected using the objective correction ring. Subsequently, grating block adjustment was performed to find the perfect focus for the laser beam interference at the focal plane. This process was automatically run by the Nikon software. To align both channels for parallel dual-color experiments NIS-Elements Two-CAM registration was used taking the same Tetraspec bead samples. Error estimation in x, y and z-axes caused by chromatic abbreviations was done by imaging multiple z-stacks of Tetraspec beads. The exact localizations of the beads in both colors were determined with sub-pixel accuracy using the imageJ plugin Quickpalm. For each bead, the x and y abbreviation was calculated and defined as a vector. The relative error of chromatic aberration was determined by the vector product of each bead with its nearest neighbor normalized to the distance of the beads. The alignment of both channels along the z-axes was done by selecting each bead in a 320 × 320 nm square, projecting it on the x–z axes, and determining the maximum intensity in both channels with sub-pixel accuracy using a Gaussian-fit in ImageJ (Supplementary fig. 5). Cells were transiently transfected with the respective constructs or siRNAs and stained, if necessary, with Mitotracker Green/ FM (MTG), Mitotracker Red/FM (MTR), or Mitotracker Red CMXROS (MTR-CMX) (Invitrogen, Thermo Fischer Scientific, Vienna, Austria). Simultaneous dual channel 3D-SIM images were acquired 90 s after placing cells in 2CaB. Experiments with MTG-stained mitochondria were not possible with the dual bandpass filter setup because of fluorescence bleed through into the red channel. Sequential dual-color 3D-SIM images were performed instead. To analyze for z-axes chromatic dispersion Hela cells were stained with MTR and MTG simultaneous and z-stacks with 60 nm increments using the dual-SIM system were acquired. The AUC (area under the curve) was measured in z-projections of single mitochondria and using linear interpolation for both labels the z-position was determined at which the AUC/2 is reached.

**Cell culture.** HeLa (ATCC-CCL-2.2), HEK (ATCC-CRL-1573) and MCF-7 (CLS-300273) cells were seeded on 1.5 H high-precision glass cover slips (Marienfeld-Superior) and cultured in DMEM (D5523, Sigma-Aldrich, Darmstadt, Germany) containing 10% FCS (Gibco, Thermo Fisher Scientific), penicillin (100 U/ml), streptomycin (100 μg/ml) and amphotericin B (1.25 μg/ml) (all Gibco) in a humidified incubator (37 °C, 5% $CO_2$/95% air). For A549 cells (CLS-300114) instead of DMEM a 1:1 DMEM and Hams F12 (Gibco, Thermo Fisher Scientific) medium mix with 10% FCS was used. Cell lines were regularly tested for myco-plasma contamination (negative). Origin of cells was confirmed by STR-profiling by the cell culture facility of ZMF (Graz). MCF-7 and HEK-293 (both ICLAC register) cells were used to prove MCU-rearrangement upon histamine in non-excitable cells that are frequently used in the respective research area.

**Cloning of constructs.** For generating mCherry or YFP-tagged constructs, the respective coding sequences were subcloned into backbones already containing the sequences for the FPs[51]. Tom22 was fused to N-terminal mCherry (mCherry-Tom22), and, cloned into a pcDNA3.1(−) backbone flanked by ClaI and HindIII restriction sites. All other constructs were fused to C-terminal mCherry (EMRE-mCherry, UCP2-mCherry, MCU-mCherry) and subcloned into a pcDNA3.1 (+) backbone. EMRE and MCU are flanked by HindIII and EcoRI, UCP2 by KpnI and EcoRI and ΔMICU1 by BamHI and EcoRI. For SIM measurements the plasmids encoding for either wild-type MICU1, MICU1 R455K (MICU1-K), MICU1 R455F (MICU1-F) or EF hand mutated MICU1 (MICU1-EF) were C-terminally fused to a citrine (YFP) as described before[12,21]. MICU1 lacking its C-terminal domain (MICU1ΔC) was fused in frame to the same YFP via BamHI and EcoRI restriction sites in a pcDNA3(+) vector using the primers MICU1 pos. 1 forward 5′-CCCGGATCCATGTTTCGTCTGAACTCACT-3′ and MICU1 pos. 1332 reverse 5′-GGTTGAATTCCATCAGCCGTTGCTTCATGAT-3′. MICU1$^{1-70}$ and MICU1$^{1-140}$ were inserted in frame via XhoI/EcoRI in a pcDNA3.1(−) vector containing a circular permutated enhanced green fluorescent protein (cpEGFP)[52] with the primers as follows: MICU1 pos. 1 forward; MICU1 pos. 210 reverse; MICU1 pos. 420 reverse. For sub-organelle-specific $Ca^{2+}$ measurements within the mitochondrial cristae (MC) a mutated version of reactive oxygen species modulator 1 (ROMO1) was synthesized by exchanging potential ROS modulating cysteines at pos. 15, 27 and 42 to serines from General Biosystems Inc. (Morrisville, NC, USA). ROMO1 and the IMS specific targeting sequence of MICU1$^{1-140}$ were then fused to a ratiometric GEMGeCO1 (Addgene; Cambridge, MA, USA) via XhoI/BamHI in a pcDNA3.1(−) with the primers ROMO1 pos. 1 forward; ROMO1 position 255 reverse; MICU1 pos. 1 forward and MICU1 position 420 reverse. For immuno-histochemistry of MCU-mCherry and EMRE-mCherry in frame with a myc-HIS-tag we constructed the myc-HIS-tag to the C-terminal end of the mCherry in a series of PCR amplifications by using overlapping reverse primers. The resulting mCherry-myc-His-tag fragment was then subcloned in fusion with either MCU or EMRE via EcoRI and HindIII restriction sites. All respective primers can be found in the Supplementary Table 1.

**Generation of UCP-KO and EMRE-KO cells.** For generating UCP2-KO or EMRE-KO cells we ordered pcDNA3.3-Cas9-2A-eGFP and pGS-U6-gUCP2/pGS-U6-gEMRE from Genscript (Thermo Fischer Scientific). The gRNA sequence of UCP2 is 5′-CCCAGTACCGCGGTGTGAT-3′ and for EMRE 5′-GCTAGTATTGG-CACCCGTC-3′. For homologous recombination in case of UCP2 we cloned an 800-bp upstream fragment (HindIII/EcoRI) and an 800-bp downstream fragment (Bsm I) of the genomic UCP2 cutting side by Cas9 into pcDNA3.1 flanking neomycin cassette. All three plasmids were co-transfected into HeLa cells in a ratio of 1:1:1. After 48 h, GFP-positive cells were single sorted into 96-well plates. Clones were further cultivated and selected with G-418 (Sigma-Aldrich) for two weeks in 12-well plates. Nearly 50 clones were analyzed with PCR. To check if each allele of 5 clones was mutated, we made bacterial sub-clones of amplified PCR fragments and sequenced them using the Sanger method (Microsynth, Balgach, Switzerland).

For generation of EMRE-KO, HeLa cells were cotransfected with pcDNA3.3-Cas9-2A-eGFP and pGS-U6-gEMRE. Next day we made single-cell sorting of GFP-positive cells into 96-well plates. Clones were further cultivated and analyzed with PCR. To check if each allele was mutated, we made bacterial sub-clones of amplified PCR fragments and sequenced them with the Sanger method (Microsynth, Balgach, Switzerland).

**Mitochondria staining.** Cells were washed once with loading-buffer containing in mM: 2 $CaCl_2$, 135 NaCl, 5 KCl, 1 $MgCl_2$, 1 HEPES, 2.6 $NaHCO_3$, 0.44 $KH_2PO_4$, 0.34 $Na_2HPO_4$, 10 D-glucose (Roth), 0.1% vitamins, 0.2% essential amino acids and 1% penicillin/streptomycin (Gibco) at pH 7.4. Cells were incubated in loading-buffer containing 0.5 μM MTG, 0.5 μM MTR, or 0.05 μM MTR-CMX (Invitrogen) for 40 min or 10 min, respectively. Thereafter, cells were washed once with loading buffer and imaged in 2CaB.

**Transfection and siRNA treatment.** HeLa cells were grown under standard culture conditions until 50% confluence was reached, transfected in DMEM (without FCS and antibiotics) with 1.5 μg/ml plasmids or 100 nM siRNA using 2.5 μg/ml TransFast transfection reagent (Promega, Madison, WI, USA)[10]. After 24 h, the medium was replaced with DMEM containing 10% FCS and 1% peni-cillin/streptomycin and kept for further 24 h before experiments. The specific siRNAs (Microsynth) used in this study are listed in Supplementary Table 1.

**Cytochrome c staining.** After staining the cells with MTR-CMX cells were washed with PBS, fixed with 3.7% paraformaldehyde in PBS for 15 min and permeabilized with 0.25% (v/v) Triton X-100/PBS for 15 min at RT. Following permeabilization, cells were washed twice with PBS and nonspecific absorption was blocked using UltraVision blocking reagent (Thermo Fisher Scientific) for 10 min at RT. Then, cells were incubated at 4 °C overnight with anti-Cytochrome c antibody (Cell Signaling, # 12963; 1:100 in Antibody Diluent, Dako). After washing with PBS, Alexa488-labeled goat-anti-mouse IgG (Invitrogen, # A11001; 1:300 in Antibody Diluent) was used as secondary antibody (60 min, RT). Slides were stored in 1% (w/ v) BSA/PBS until further use.

**Sub-mitochondrial localization of UCP2, MCU, EMRE and TOM22.** Cells transfected with EMRE-mCherry, MCU-mCherry, UCP2-mCherry or mCherry-TOM22 were stained with MTG and imaged using 3D-SIM microscopy. Images were background subtracted with an ImageJ Plugin (Mosaic Suite, background substractor, NIH). The width of mitochondria was measured using the full width at half maximum (FWHM). Line plots were taken for the images in ImageJ and the FWHM was determined.

**Assigning MICU1 localization to inner mitochondrial membrane.** Dual-color 3D-SIM images of MICU1-YFP and mCherry-TOM22 transfected cells in 2CaB were background subtracted with an ImageJ Plugin (Mosaic Suite, background substractor, NIH). Blind line plots of mitochondrial cross-sections within the mCherry-TOM22 label were done. The outer mitochondria membrane can be modeled as a hollow tube that, if stained, displays a double line structure in z-projection[53]. Intensity line plots of MICU1-YFP and mCherry-TOM22 were fitted to a double Gauss-function. The distance of the peak values of these fits can be interpreted as the diameter of the mitochondria and were determined for both channels. By bisecting the difference of peak distances of MICU1-YFP and mCherry-TOM22 staining the distance of both labels on one side can be deter-mined. Using the hollow tube model and the Gauss fitting model, the precision for estimating the center localization of MICU1 and TOM22 distributions can be increased beyond the actual resolution of SIM microscopy as long as the mito-chondrial double line structure in images or the double peak structure of mito-chondrial cross-section intensity plots is present. As control cells were transfected with MICU1 YFP and –mCherry. For image analysis the freeware program ImageJ (NIH, MA, USA) was used.

**MICU1 time-lapse imaging during mitochondria depolarization.** Time-lapse imaging of MICU1 cristae relocalization was performed with 30 s time increments over 300 s in 2CaB supplemented with 2 μM oligomycin A (Sigma-Aldrich). 90 s

after acquisition started, antimycin A was added into the live-cell chamber to a final concentration of 4 μM.

**Analysis of protein sub-mitochondrial localization.** Dual-color images were split into two separate channels, background corrected with an ImageJ Plugin (Mosaic Suite, background subtractor, NIH) and further described as object and reference channels. The reference channel was Otzu auto thresholded[54] and further single dilated (iterations = 1–4, count = 1) and single treated with an erosion algorithm (iterations = 2–6, count = 1). Erosion, dilation iterations and counts were changed according to the experiment to get the highest sensitivity possible for each experimental set. More detailed information about the parameters used and comparability between the experiments is listed elsewhere (Supplementary Table 2). Different settings of erosion and dilation parameters were necessary to extract the sub-mitochondrial localization using different reference markers with variable mitochondrial localization patterns. In case of MICU1-YFP as the reference after auto thresholding, a fill holes step was added. Pixel-wise subtraction of the erosion reference of the dilated reference image yields in a hollow structure. The hollow structure was further used as a template to determine the fluorescence intensity in the IBM while the erosion reference served as template for the bulk or cristae fluorescence intensity. The measured fluorescence intensities were normalized to the area covered by the cristae and IBM template. The ratio of IBM/CM mean intensity is a value to estimate changes of the object label distribution inside a mitochondrion and termed IBM association index. The higher the ratio value the higher the distribution of protein label in the IBM. For image analysis the freeware program ImageJ was used.

**Electron microscopy and analysis of cristae junction width.** Cells were washed in PBS, fixed with 2.5% glutardialdehyde and 2% formaldehyde in a buffered solution and postfixed in 1% osmium tetraoxide that had been reduced with 1% potassiumhexacyanoferrate[34]. The cells were dehydrated in an ascending ethanol series, embedded in TAAB embedding resin, and sectioned on a Leica Ultracut 7 ultramicrotome using a Diatome diamond knife. The sections were counter stained using platinum blue (IBIlabs) and lead citrate (Leica) and visualized in an FEI Tecnai 20 transmission electron microscope. They were photographed at ×27,000 magnification with a Gatan ultrascan 1000 camera. To quantitatively analyze the electron microscopically images a line was drawn in imageJ manually into the cristae starting with the cristae junction proceeding into the cristae volume as far as the image quality was sufficient for image quantification, the curvature of the cristae was not crossing the line or a maximum of 130-nm length was reached. Along the drawn line every 2 nm orthogonally line plots with a width of 10 nm were measured with an ImageJ macro starting from the CJ. Each line plot was halved and the position of the minimal intensity was determined for both sides. The distance of both minimal intensities was set as the distance between opposing sides of the cristae.

**Morphological analysis of mitochondria imaged with SIM.** Single 3D-SIM and time-lapsed images of EMRE-mCherry, MTR, MTG, or MICU1-YFP were used for morphological analysis over time. Images were background corrected with an ImageJ Plugin (Mosaic Suite, background subtractor, NIH) and a binarization was done using a Yen auto threshold[55]. The ImageJ particle analyzer was used to extract the mitochondrial count ($c$), area ($a$), perimeter ($p$), minor ($x$) and major ($y$) axes of the mitochondria. Aspect ratio (AR) was determined as $AR = \frac{y}{x}$.

The form factor (FF) was determined as follows: $FF = \frac{p^2}{4\pi \cdot a}$. For image analysis the freeware program ImageJ was used.

**3D-morphological analysis of mitochondria.** $z$-stacks of mitochondria with 0.2-μm increments were imaged with a Zeiss Observer Z.1 inverted microscope equipped with a Yokogawa CSU-X1 Nipkow spinning disk system, a piezoelectric $z$-axes motorized stage (CRWG3-200; Nippon Thompson Co., Ltd., Tokyo, Japan), and a CoolSNAP HQ2 CCD Camera (Photometrics). The image stacks were deconvoluted using blind deconvolution (NIS-Elements, Nikon, Austria). Morphology parameters were measured automatically with a custom made ImageJ macro using the following procedure. An additional background subtraction using the rolling ball method was used to further increase contrast for later analysis. Both a global auto Otsu threshold using the stack histogram as well as a local auto Otsu threshold (radium of 640 nm) using the single slice histogram were applied to the stack and merged. The ImageJ plugin 3D manager was used to segment the binarized mitochondria. With the plugin 3D Geometrical Measure the mitochondrial volume and surface were determined. The Plugin 3D Ellipsoid Fitting generated an ellipsoid fit of the mitochondria to measure elongation and flatness parameters.

**CM kinetics quantification.** HeLa cells transfected with ER-RFP and stained with MTG were recorded with live dual-color SIM imaging. Analysis of the CM and IBM kinetics was done as described elsewhere[28]. In short, using a combination of local and global automated thresholding of the MTG staining the CM-membrane kinetics were quantified by measuring the CM alterations pre frame within the mitochondria. The overlap of ER-RFP and MTG staining was defined as MAMs.

The CM kinetics were determined inside the MAMs using the overlap of ER and mitochondrial staining as a mask.

**Mitochondrial shape descriptors and co-localization.** HeLa cells transfected with ER-RFP and stained with MTG were recorded with live dual-color SIM imaging. The mitochondrial branching was measured by determining the area weighted FF and mitochondrial size using ImageJ[28]. For co-localization studies the ImageJ plugin coloc2 was used to measure the Pearson coefficient.

**Cell viability and apoptosis assay.** Twenty-four hours after transfection in 10 cm dishes, cells were seeded in 96-well plates at a density of 5000 cells per well. Cell viability was measured using CellTiter-Blue assay (Promega) and apoptotic caspase activity via Caspase-Glo3/7 assay (Promega) following the standard protocols[56].

**Live cell Fura-2, TMRM, and mitochondrial Ca²⁺ measurements.** Measurements were performed on an inverted and advanced fluorescent microscope with a motorized sample stage (Till Photonics, Graefling, Germany). The probes were excited via a polychrome V (Till Photonics), and emission was visualized using a ×40 objective (alpha Plan Fluar 40, Zeiss, Goettingen, Germany), and a charge-coupled device camera (AVT Stringray F145B, Allied Vision Technologies, Stadtroda, Germany). Fura-2 (Invitrogen) was alternately excited at 340 and 380 nm, and emissions were captured at 515 nm (495dcxru; Omega Optical, Brattleboro, VT, USA). TMRM (cat: T668; Invitrogen) was excited at 550 nm, and emissions were captured at 600 nm (59004; Chroma, Bellows Falls, VT, USA). Mitochondrial Ca²⁺ FRET measurements with 4mtD3cpv were done using 430 nm and 480 nm excitation and emission was captured at 480 and 535 nm (69008; Chroma), respectively[12]. For control and acquisition, the live acquisition 2.0.0.12 software (Till Photonics, Munich, Germany) was used. For Fura-2 imaging cells were incubated in loading-buffer containing 3.3 μM Fura-2 for 40 min[56]. TMRM loading was done likewise for 40 min with a TMRM concentration of 100 nM[57]. For mitochondrial Ca²⁺ measurement cells were transfected with the 4mtD3cpv-Plasmid[9]. All recordings were background subtracted using a background ROI and corrected for bleaching using an exponential decay fit. Mitochondrial targeted R-GECO was used to measure mitochondrial Ca²⁺ using the single-channel SIM-setup in wide field mode. Time-lapse images were background subtracted using a background ROI and corrected for bleaching using an exponential decay fit.

**Live cell measurement of IMS and CL Ca²⁺.** Ratiometric green emitting genetically encoded Ca²⁺ indicators for optical imaging (GEMGeCO1) fused to either ROMO1 or MICU1¹⁻¹⁴⁰ was excited at 430 nm and emissions were collected by using a conventional CFP/YFP dichroic filter at 480 and 535 nm. The resulting ratio was calculated by the division of the fluorescence intensity at 480 nm through 535 nm. CARGeCO1-based Ca²⁺ indicators (cyto-CARGeCO1 and mt-CAR-GeCO1) were excited at 575 nm and emitted at 600 nm. For simultaneous measurements, GEM- and CARGeCO1 targeted sensors were alternately excited for 400 ms each at 430 and 575 nm. Emissions derived from both sensors were taken in 3 s intervals. Alternatively, we used an ultrafast switching mode where both Ca²⁺ indicators were excited for 150 ms in a 310 ms interval. All recordings were background subtracted using a background ROI and corrected for bleaching using an extrapolation of an exponential decay fit. The maximal slope values and duration of lag time evaluated from the onset of cytosolic Ca²⁺ rise to the maximal slope were calculated.

**mRNA Isolation and real-time PCR.** Total RNA was isolated according to manufacturer's protocol using the PEQLAB total RNA isolation kit (Peqlab, Erlangen, Germany). cDNA was generated using a high-capacity cDNA reverse transcription kit (Applied Biosystems, Foster City, CA) and a thermal cycler (Peqlab). Knockdown and overexpression efficiency of MICU1 was examined by RT-PCR using a QuantiFast SYBR Green RT-PCR kit (Qiagen, Hilden, Germany) and a LightCycler 480 (Roche Diagnostics, Vienna, Austria). Relative expression of the gene of interest was normalized to GAPDH, as a reference gene. Primers for real-time PCR were designed by us and obtained from Invitrogen (Vienna, Austria). The respective primer sequences used can be found elsewhere (Supplementary Table. 1).

**Co-immunoprecipitation.** HeLa or HEK cells were transfected with MICU1-FLAG alone or in combination with MCU-mCherry-HIS or EMRE-mCherry-HIS constructs. The transfection process was repeated three times over a period of 3 days. Transfected cells were lysed in 0.7 ml lysis buffer (100 mM NaCl, 20 mM Tris, 1 mM EGTA, 5 mM n-Dodecyl β-D-maltoside, pH 7.5-HCl) supplemented with protease inhibitors (10 μM phenylmethylsulfonyl fluoride, and 1 μg/ml each aprotinin, leupeptin, and pepstatin). After a short freeze-thaw cycle lysates were centrifuged (12,000×g, 10 min) and 60 μl of supernatants were removed for protein determination using a BCA assay (Thermo-Fisher) and total cell lysate analyses (input). The rest of the lysates (~1.2 mg total protein of HeLa cells and ~2 mg of HEK cells) were incubated with 30 μl of anti-Flag M2 affinity gel (Sigma-Aldrich; A2220-1ML; beads were prepared as recommended by the manufacturer) for 30 min on a rotary shaker. Then, beads were collected by centrifugation (8000 × g,

30 s) and washed three times with TBS. All steps were performed on ice or at 4 °C. Elution of FLAG-fusion proteins was performed in 50 μl of 2× Lämmli sample buffer with 5% β-mercaptoethanol and boiling for 10 min. For immunoblotting, whole cell lysates (input, 120 μg of total protein) and Co-IP samples were subjected to SDS-PAGE. PVDF membranes were probed at 4 °C overnight with primary monoclonal mouse anti-6x His-Tag antibody (Thermo Fisher; 1:1000; MA1-21315). To avoid interference from denatured heavy and light chain IgG, the HRP-conjugated immunoblot reagent Veriblot (Abcam; 1:000; ab131366) was used for detection of co-immunoprecipitated proteins. For detection of protein expression in input samples, HRP-conjugated goat-anti-mouse secondary antibody (Thermo Fisher, 1:2500, 31430) was used. Immunoreactive bands were visualized using Immobilon Western HRP Substrate (Thermo Fisher) and the chemiluminescence detection system ChemiDoc (Bio-Rad).

**Western Blot for MICU1 degradation analysis.** HeLa cells plated in 6-well plates were treated with 2 μM antimycin A and 4 μM oligomycin A for 2 and 10 min. For immunoblotting, whole cell lysates were prepared with radio immunoprecipitation assay (RIPA) buffer and 100 μg of total protein was subjected to SDS-PAGE. PVDF membranes were probed with primary polyclonal rabbit anti-MICU1 (Cell Signaling; 1:1000; 12524 S) and HRP-conjugated goat-anti-rabbit (Sigma; 1:1000; 12–348) antibodies. For normalization, membranes were stripped and re-probed with primary antibody against β-actin (Sigma; 1:1000; A5316). Immunoreactive bands were visualized using Immobilon Western HRP Substrate (Thermo Fisher) and the chemiluminescence detection system ChemiDoc (Bio-Rad).

**Statistical analysis and reproducibility.** Each exact n value and the number of independent experiments is indicated in each figure legend. Statistical analysis was performed using the GraphPad Prism software version 5.04 (GraphPad Software, San Diego, CA, USA) or Microsoft Excel (Microsoft Office 2013). Analysis of variance (ANOVA) with Bonferroni post hoc test and $t$-test were used for evaluation of the statistical significance. All Box-plots show minimum to maximum values. The central line is the median with boxes extending to 25 and 75% and the whiskers encompass all data. $P < 0.05$ was defined to be significant. At least three different experiments on three different days performed in at least triplicates have been performed for each experimental setup.

**Reporting Summary.** Further information on research design is available in the Nature Research Reporting Summary linked to this article.

## Data availability

The data that support the findings of this study are available from the authors on reasonable request, see author contributions for specific data sets. All source data for the preparation of this manuscript are available from the authors on request. The source data underlying Figs. 1d, 2d, e, f, 3a, b, 4d, f, 5a, c, d, f-h, 6b, d, 7b, d, 8a and c and supplementary figures 1, 3–5, 7, 8, 10–24 and 26–36 are provided as a Source Data file.

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

## Acknowledgements

The authors wish to thank Anna Schreilechner, BSc and Sandra Blass, BSc and Elisabeth Pritz, BSc for their excellent technical assistance. This work was funded by the Austrian Science Funds (FWF, DKplus W 1226-B18 and P 28529-B27). B.G. is supported by Nikon Austria, and he and C.K. are doctoral fellows within the doctoral program *Metabolic and Cardiovascular Disease* (MCD) (FWF, DKplus W 1226-B18) at the Medical University of Graz. H.B. is a fellow of the doctoral program *Molecular Medicine* (MolMed) at the Medical University of Graz. The SIM equipment is part of the Nikon-*Center of Excellence*, Graz that is supported by the Austrian infrastructure program 2013/2014, Nikon Austria Inc. and BioTechMed.

## Author contributions

B.G., C.K., C.T.M.-S., E.E., H.B., E.B., W.S. and M.W.-W. performed experimental work and FRET measurements, M.W.-W., C.K., and E.E. cloned the constructs, R.R. was responsible for cell culture and transfection and production of knockout cells, B.G., G.L. and S.R. planned and performed the electron microscopy experiments, B.G. performed super-resolution microscopy and image data analyses and created Fig. 1b, d, 4g, 5b, and 5e, W.F.G. together with R.M. supervised the research and project planning, performed data interpretation and prepared the manuscript. All authors discussed the results and implications and commented on the manuscript at all stages.

## Additional information

**Competing interests:** The authors declare no competing interests.

