## [Peer Review File · Nature Communications]

Reviewers' comments:

Reviewer #1 (Remarks to the Author):

The mitochondrial calcium uniporter complex resides within the organelle's inner membrane, which includes both an inner boundary membrane (IBM) and cristae membrane. Here, the authors show that the regulatory component MICU1 preferentially localizes to the IBM while the uniporter's pore-forming components exist throughout the inner membrane, but to the IBM during calcium signaling. The authors also show that MICU1 is important for cristae junction stabilization (and in fact, when over-expressed, can even compensate for loss of OPA1). These observations are exciting likely to be of broad interest in the field. However, several points need to be addressed before the story is ready for publication.

Major Points:

(1) The IBM association index is confusing. What value is considered IBM and what is homogeneously localized across the IMM? The metric / threshold varies significantly between figures. Either the authors do not consider this metric quantitative (which itself would be a problem) or a further explanation is needed. (For example: -In Figure 2d, the authors are describing the IBM association of at least 0.7 for MICU1 with Antimycin to be "diffusion of MICU1 into the CM", whereas the same value (or lower on average) in Figure 2g for MICU1 1-140 is described as "mimicked the IBM location of wild type MICU1". - Likewise, an IBM association difference of ~ 0.15 between MICU1 1-140 and MICU1 1-70 is interpreted as a difference in sub-mitochondrial localization of these mutants—is there any evidence that such a small difference should be interpreted so boldly? - The authors suggest that an exclusive localization to inside the cristae for cytochrome c gives an IBM association of ~ 0.35 , whereas spread out throughout the IMS is ~ 0.4 . - For MCU (Figure 8), this metric gets even more confusing where ~ 1 is not considered IBM (for MICU1 1 seems to be considered exclusively IBM in Fig 2 and even 0.7 elsewhere), whereas ~ 1.1 is IBM. Please be quantitative, select a threshold, and use it throughout.)

(2) The authors show that MICU1 KD widens the CJ, but is this a direct or indirect effect of MICU1? For example, MICU1 loss can lead to chronic mito calcium overload, which may itself affect the CJ. While the authors have elsewhere reported that CM-dynamics was independent of acute matrix calcium signal, chronic calcium overload may be different. Chronic calcium overload vs. MICU1 direct CJ function would be easy to distinguish in the authors' set up by completely knocking out MCU or EMRE in the MICU1 KD cells. If these assays can be performed in a way that allows Ru360 to access mitochondria, then pharmacologically targeting the uniporter may be an option as well. Separately, the authors show that MICU1 KD disrupts the mitochondrial membrane potential (figure 5a) and that changes in the membrane potential impact MICU1 localization (figure 2). Does treating with a protonophore (CCCP) or valinomycin have any effect by itself on the CJ? Does every mito protein KD that impacts membrane potential show the same result in this assay? Such experiments are required to evaluate a direct versus indirect effect.

(3) EMRE with a C-terminal fluorescent protein tag has not been shown to be functional in the literature before, so it is necessary to show that this fusion is functional. EMRE has been shown to have dual functions within the uniporter: (1) it is required for calcium transport through the channel and (2) it is required for MICU1 interaction with the channel. In order for the results with EMRE C-terminally tagged with a fluorescent protein to be interpretable, the authors need to show whether or not this fusion protein retains both functions. If their fusion protein is not fully functional, it may be best to focus on results only with MCU and MICU1.

(4) It appears that every experiment has been performed in Hela cells. It is important to know whether this phenomenon is robust and extends to other cell types. What would be particularly useful is to examine the skeletal muscle from either human MICU1 myopathy patients or the mouse model of MICU1 deficiency and evaluate whether this phenomenon extends there. At the very least the authors

ought to test another cell type to ensure this is a robust result.

(5) The term "MCUC" for the uniporter complex is very confusing now that Docampo et al have identified additional MCU homologs (including MCUC). The authors could use the term "uniplex" or simply drop an abbreviation and use the term "uniporter complex" to avoid confusion.

Minor points:

(1) In Figure 2a, it is clear in the model that the authors do not think MICU1 has a TMD and there is no evidence for MICU1 having a transmembrane domain. Perhaps the authors mean mitochondrial targeting sequence?

(2) It has been shown that MICU1 interaction with MCUC is more stable in the absence of calcium (Petrungaro et al 2015). This contradicts the authors' model in which calcium increases complex assembly. Please provide discussion on this point.

(3) The authors say in the discussion that Sancak et al. 2013 reported that EMRE binds MCU but not MICU1, which was not actually tested in this reference.

(4) The authors reference Wang et al 2014 for showing that MICU1 oligomeric state changes with calcium. In fact, they did not show this: they show that MICU1 Δ C forms a dimer in the presence or absence of calcium and MICU1 forms a hexamer in the absence of calcium and either a higher order oligomer or just aggregated protein under these conditions in vitro in the presence of calcium. Similarly, the authors state that MICU1- Δ C mutant "does not oligomerize with wild-type MICU1" on page 5—this has not been shown, including in the Wang et al reference provided, and is not important to the conclusions. This statement should be removed.

Reviewer #2 (Remarks to the Author):

In this manuscript, authors report several new discoveries on the biology of MICU1. They show the localization of MICU1 at the mitochondrial inner boundary membrane (IBM) and describe domains responsible for such localization. They also show the importance of MICU1 in mitochondrial cristae width, an effect almost comparable to that of OPA1. At the end of the paper authors describe the importance of UCP2 and MICU1 methylation for MCU anchoring at the IBM. In its present form, the manuscript presents provocative and potentially important data. However, some experiments are required to support the conclusion of the paper.

Our main concern lies on using mostly microscopy data without any biochemical data supporting them.

1) First outer membrane diameter over 500nm as shown in Fig.1, Fig.6 and 7 are double the size of what was measured by other groups using high resolution microscopy (Jakobs and Wurm 2014 in Current Opinion in Chemical Biology Volume 20, June 2014, Pages 9-15 describe outer membrane diameter to be 280nm (STED microscopy) and Plecítá-Hlavatá et al showed it to be 250nm by PALM microscopy (FASEB J. 2016 May;30(5):1941-57.), suggesting that perhaps they are stressed and acquire a doughnut like shape. Most of the mitochondria are becoming swollen (they have a hole in mitotracker signal) and for example mitochondria after histamine treatment seems to have higher diameter than untreated. This should be measured and stated in the manuscript.

2) Is there any reference or clear experiment confirming the transmembrane domain in MICU1? So far there are only papers claiming cardiolipin association but not clear transmembrane localization (Kramer et al, EMBO Rep. 2017 Aug;18(8):1397-1411.). because of this discrepancy, authors shall verify if cardiolipin deficiency (eg, tafazzin deletion) displaces MICU1

3) Another problem lies in the exact localization of MICU1. Authors claim intermembrane space localization but there is another publication showing mitochondrial matrix localization (Hoffman et al

2013, Cell Reports, Volume 5, Issue 6, 26 December 2013, Pages 1576-1588). Therefore, I would suggest more biochemical experiments to support the authors' conclusion, like standard carbonate extraction and proteinase K treatment to reveal membrane localization.

4) According to the supplementary figures, all of the tested proteins are 60X overexpressed in comparison to wt levels of proteins. This is a major concern. Could authors come to same conclusions using for example immunocytochemistry against native MICU1? What if overexpression of MICU1 just leads to artifacts? Maybe MICU1 overexpression is widening mitochondria. Did authors compare thickness of mitochondria (or IBM) in controls and cells overexpressing MICU1? All of the differences between wt and overexpression should be discussed in the manuscript. For example, what about cristae morphology? Are cristae tightened in OE cells, if upon ablation they are widened?

5) In Fig.2 authors show a change in localization of MICU1 upon membrane potential depletion. However, this experiment is raising many questions. Is MICU1 localization changed because of proteolytic cleavage or of changes in oligomerization? This questions can be answered by SDS and BNPAGE electrophoresis.

6) How does Δ C-MICU1-YFP interact with WT MICU1, if the C-helix was deleted? Can authors show this on native PAGE?

7) In the second paragraph on page 5 authors conclude that "The contribution of OPA1 on the integrity of the CJ was verified with transmission electron microscopy (TEM) that revealed a widening of the CJ by the knockdown of OPA1 (Fig. 4a,b&d) and, thus, explains the movement of the Δ C-mutant of MICU1 into the CM". How does OPA KD explain the movement of the Δ C-mutant of MICU1 into the CM? Could authors better explain their conclusion?

8) What is the physiological significance of having MICU1 in the IBM if MCU is in the cristae? What is the relationship between MICU1 in the IBM and mPTP? Can MICU1 at the IBM have any effect on mPTP opening?

Reviewer #3 (Remarks to the Author):

Gottschalk et al present data relating to distribution of MICU1 and a number of other factors mostly in dual colour live cell SIM imaging with some data from electron microscopy. Although the paper appears to show some interesting results, some of the presented data does not appear to support the conclusion they come to and their discussion makes following their reasoning difficult.

Main issues:

Figure 1) and text "we estimated a distance of 22 ± 4 nm between the distributions of MICU1 and TOM22"

Is this results estimated or measured? Whichever, it is an extremely bold claim, given the data is a) in different colours and so subject to dispersion and chromatic aberration, and b) their image resolution, although not stated or measured is likely to be somewhere around 120-140 nm and they claim to have a accuracy of < 5nm after making just 24 measurements.

I would be particularly worried that Z plane misalignment would image different regions in the different colours leading to possibly systematic errors in such a small sample size. Even if the imaging system is perfectly aligned, dispersion with the live cell will affect the Z magnification and hence image alignment between channels with

imaging depth.

Figure 2b) The selection of the marked region seems to provide the answer that the authors want to find. Moving the selected line scan roughly 500 nm NW would produce a result similar to T=270 at both time points, whereas moving a similar distance SE would produce the T=0 result at both time points. The data selection seems extremely biased.

I am also surprised by the significance in plot 2g, however since you don't state what the box and whiskers represent on the plot it is impossible to tell if this significance is likely to be real or not.

In supplementary figure 2, the IMM/MTG staining appears morphologically quite different in TOM22 compared to all the others. The mitochondria seems to be much thinner and less heavily stained for IMM/MTG

It seems that figure 3 has a similar shift in the morphology with much thinner mitochondria in the delC-MICU1 images whether there is a wildtype knockdown or not.

Supplementary Fig 7, movement is shown in arbitrary units, how is this sensible. Also how is this movement defined? Mean Square Displacement? $\mu\text{m/s}$?

The text states "after stimulation with histamine, MCU joins MICU1 to exclusively localise in the IBM." This just isn't true. Yes it is more in the same places as the MICU1 but definitely not exclusively, look at the blown up plot in the bottom right panel of 6a, there are significant purple areas which are not green.

Fig 6b shows box and whisker plots for 4 conditions, with one being marked as significant, but what is this test against? I assume that it is the +hist control but you don't say this? How do you account for the lower IBM association index in the -hist experiment?

In discussing this figure you state "These data indicate...active MCUC dynamically assembles because MICU1 traps MCU in the IBM", I think this is a proposed mechanism suitable for the discussion but not in the results section. Also supplemental fig 13 is not strong evidence for this, the IBM association factor goes up by only about 10%.

Page 8: the references to sup fig 15,16 are wrong, and refer to data which isn't there, and sup fig 16 is missing.

Discussion. The discussion is hard to follow with a multitude of similar acronyms in quick succession (CJ,CM,CL) for those not intimately familiar with these it is very hard to follow the logic of what the authors are trying to describe. I think they also overstate their case in several places "localisation of MICU1 depends on the proteins ability of electrostatic repulsive power." All the authors show is that the 70-140 residue region is required.

They also state that

"Our findings that knockdown of OPA1 did not change the exclusive IBM location of wild-type MICU1 but facilitated the redistribution of Δ C-MICU1 into the CM even in the presence of endogenous MICU1 suggest a model in which the sub-mitochondrial localization of MICU1 to the IBM is dependent on the stability of the CJ."

I don't follow the argument as to why the re-localisation is then dependant on CJ stability?

Overall the discussion is extremely hard to follow and could benefit from a diagram showing how the proteins and CJ structure interact in their proposed mechanism.

A large number of experiments depend upon the use of siRNA techniques, and yet limited no controls to demonstrate that the effects seen are not off target results. siRNA experiments need to have scrambled negative controls and two different siRNAs to be reliably interpreted. The one experiment which seems to have a negative control is shown in Sup fig 8, but there is no information about the "control si" used.

Several of the image analyse techniques are said to include background subtraction, but how this is achieved is only stated in the Morphological analysis of mitochondria over time section, with no suggestion that this technique was used elsewhere.

Many of the analysis routines use an Otsu auto threshold, whereas the morphological analysis uses a Yen auto threshold, why?

The extensive use of box and whisker plots is nice, however the authors must specify exactly what the boxes and whiskers represent, otherwise they are meaningless. There are also situations where it is not clear which values are being compared to test for statistical significance.

Minor issues.

Supplemental figure 5, it is hard to follow when mutants are called MICU1-R455F-YFP in the text and MICU1-F in the figure.

Response to the Referees

We thank all referees very much for their valuable, fair and very helpful comments that helped us to revise our manuscript accordingly. In particular, the authors truly appreciate the time and efforts all referee spent to help us in improving our work. Below, please find our point-to-point response indicating how we addressed the issues raised (blue):

Reviewer #1:

The mitochondrial calcium uniporter complex resides within the organelle's inner membrane, which includes both an inner boundary membrane (IBM) and cristae membrane. Here, the authors show that the regulatory component MICU1 preferentially localizes to the IBM while the uniporter's pore-forming components exist throughout the inner membrane, but to the IBM during calcium signaling. The authors also show that MICU1 is important for cristae junction stabilization (and in fact, when over-expressed, can even compensate for loss of OPA1). These observations are exciting likely to be of broad interest in the field. However, several points need to be addressed before the story is ready for publication.

We thank the referee very much for the clear synopsis and the kind word on our work. We have addressed each point raised and hope to meet the referee's expectations.

Major Points:

(1) The IBM association index is confusing. What value is considered IBM and what is homogeneously localized across the IMM? The metric / threshold varies significantly between figures. Either the authors do not consider this metric quantitative (which itself would be a problem) or a further explanation is needed. (For example: -In Figure 2d, the authors are describing the IBM association of at least 0.7 for MICU1 with Antimycin to be "diffusion of MICU1 into the CM", whereas the same value (or lower on average) in Figure 2g for MICU1 1-140 is described as "mimicked the IBM location of wild type MICU1". - Likewise, an IBM association difference of ~ 0.15 between MICU1 1-140 and MICU1 1-70 is interpreted as a difference in sub-mitochondrial localization of these mutants—is there any evidence that such a small difference should be interpreted so boldly? - The authors suggest that an exclusive localization to inside the cristae for cytochrome c gives an IBM association of ~ 0.35 , whereas spread out throughout the IMS is ~ 0.4 . - For MCU (Figure 8), this metric gets even more confusing where ~ 1 is not considered IBM (for MICU1 1 seems to be considered exclusively IBM in Fig 2 and even 0.7 elsewhere), whereas ~ 1.1 is IBM. Please be quantitative, select a threshold, and use it throughout.)

Thank you very much for this valuable point. To address the referee's criticism, we changed the analysis setting for quantification of figure 2g to match the similar analysis in figure 3a&b. Furthermore, we included a table (**Supplementary Table 2**) listing the precise analysis setting for each set of experiments, to give an overview of which values can be directly compared to each other. Since the calculation of the IBM association factor depends on the reference labeling and imaging methodology used, not all figures and values can be directly referenced to each other. This was now mentioned in the Methods section, lines 651 to 653.

(2) The authors show that MICU1 KD widens the CJ, but is this a direct or indirect effect of MICU1? For example, MICU1 loss can lead to chronic mito calcium overload, which may itself affect the CJ. While the authors have elsewhere reported that CM-dynamics was independent of acute matrix calcium signal, chronic calcium overload may be different. Chronic calcium overload vs. MICU1 direct CJ

function would be easy to distinguish in the authors' set up by completely knocking out MCU or EMRE in the MICU1 KD cells. If these assays can be performed in a way that allows Ru360 to access mitochondria, then pharmacologically targeting the uniporter may be an option as well. Separately, the authors show that MICU1 KD disrupts the mitochondrial membrane potential (figure 5a) and that changes in the membrane potential impact MICU1 localization (figure 2). Does treating with a protonophore (CCCP) or valinomycin have any effect by itself on the CJ? Does every mito protein KD that impacts membrane potential show the same result in this assay? Such experiments are required to evaluate a direct versus indirect effect.

Thank you very much for this very helpful advice. According the referee's suggestion, we performed further electron microscopy experiments on MICU1, MCU/EMRE and MICU1/MCU/EMRE siRNA treated cells. These new results show no influence of MCU/EMRE knockdown on the CJ morphology. The results were described in the manuscript (results 189 to 194; discussion 399 to 401) and **Supplementary Figure 16**. In addition, Tufi et al. just recently showed in a preprint that MICU1-KO flies do not recover from the lethal phenotype by parallel MCU or EMRE KO, indicating an apoptotic principle of action which does not involve mitochondrial Ca^{2+} uptake as a key element (Tufi et al. 2018). In this context, the CJ bottleneck function of MICU1 might play a crucial role in maintaining the cytochrome C distribution in the cristae, and in suppressing apoptosis, as we show in **Figure 4**.

To address the referee's comment regarding the impact of a knockdown of other mitochondrial proteins of the mitochondrial uniporter complex and if they affect membrane potential, TMRM measurements were performed. These measurements did not show changes in mitochondrial membrane potential by knockout or knockdown of the MCU-complex core proteins EMRE and MCU, respectively. This is now presented in the results (lines 226 to 227) and shown as new **Supplementary Figure 20**.

(3) EMRE with a C-terminal fluorescent protein tag has not been shown to be functional in the literature before, so it is necessary to show that this fusion is functional. EMRE has been shown to have dual functions within the uniporter: (1) it is required for calcium transport through the channel and (2) it is required for MICU1 interaction with the channel. In order for the results with EMRE C-terminally tagged with a fluorescent protein to be interpretable, the authors need to show whether or not this fusion protein retains both functions. If their fusion protein is not fully functional, it may be best to focus on results only with MCU and MICU1.

Thank you very much for pointing out this important issue. To address this point, we performed mitochondrial Ca^{2+} experiments in EMRE knockout cells and functionally rescued mitochondrial Ca^{2+} uptake by an expression of the EMRE-mCherry construct. These results show that the EMRE-mCherry construct is functionally active. These new results are now mentioned lines 290 to 292 and in **Supplementary Figure 29**.

Regarding the binding of EMRE to MICU1, we performed co-immunoprecipitation experiments using MICU1-FLAG and EMRE-mCherry-HIS or MCU-mCherry-HIS constructs. Similar to the previous reports (Sancak et al. 2013; Phillips et al. 2019), we could show that EMRE-mCherry-HIS as well as MCU-mCherry-HIS binds to MICU1-FLAG. These new results are now mentioned in the result section (lines 292 to 294), in the discussion (lines 443 to 460) and presented as **Supplementary Figure 29**.

(4) It appears that every experiment has been performed in Hela cells. It is important to know whether this phenomenon is robust and extends to other cell types. What would be particularly useful is to examine the skeletal muscle from either human MICU1 myopathy patients or the mouse model of

MICU1 deficiency and evaluate whether this phenomenon extends there. At the very least the authors ought to test another cell type to ensure this is a robust result.

We are thankful you raised that point. Following the advice of this referee, we performed experiments to validate the MCU-shuttling process in non-excitabile HEK, A549 and MCF-7 cells. These results are added in the result section (lines 275 - 277), the discussion (lines 466 – 468 and 474 - 479) and as new **Supplementary Figure 27**. In addition, the mitochondrial morphology of these cells was determined and the results were added to the manuscript in lines 277 – 281 and in new **Supplementary Figure 28**.

(5) The term “MCUC” for the uniporter complex is very confusing now that Docampo et al have identified additional MCU homologs (including MCUc). The authors could use the term “uniplex” or simply drop an abbreviation and use the term “uniporter complex” to avoid confusion.

We replaced “MCUC” with “MCU-complex” throughout the entire text.

Minor points:

(1) In Figure 2a, it is clear in the model that the authors do not think MICU1 has a TMD and there is not evidence for MICU1 having a transmembrane domain. Perhaps the authors mean mitochondrial targeting sequence?

Thank you for this correction. We replaced TMD (transmembrane domain) with MTS (mitochondrial targeting signal) throughout the manuscript, figure 2a and the respective figure legend.

(2) It has been shown that MICU1 interaction with MCUC is more stable in the absence of calcium (Petrungaro et al 2015). This contradicts the authors’ model in which calcium increases complex assembly. Please provide discussion on this point.

Thank you for mentioning this important point. However, we think that the experimental setup of Petrungaro *et al.* is different from ours. While they show a reduction in MICU1/2-MCU interaction in the presence of 5 mM free Ca^{2+} , this cannot be compared directly to conditions in living cells (Petrungaro et al. 2015). Lysis of cells, even under mild conditions, disrupts the mitochondrial membrane potential, which we think is crucial for MICU1 localization to the IBM, MICU1-IMM interaction, and potentially MICU1-MCU interaction. Furthermore, only a reduction of MICU1/MICU2 heterodimer interaction with MCU was observed, while MICU1 monomers did not show a changed binding affinity to MCU under 5mM free Ca^{2+} .

(3) The authors say in the discussion that Sancak et al. 2013 reported that EMRE binds MCU but not MICU1, which was not actually tested in this reference.

We changed this passage in the manuscript (lines 442 – 443).

(4) The authors reference Wang et al 2014 for showing that MICU1 oligomeric state changes with calcium. In fact, they did not show this: they show that MICU1deltaC forms a dimer in the presence or absence of calcium and MICU1 forms a hexamer in the absence of calcium and either a higher order oligomer or just aggregated protein under these conditions in vitro in the presence of calcium. Similarly, the authors state that MICU1-delta-C mutant “does not oligomerize with wild-type MICU1” on page 5—this has not been shown, including in the Wang et al reference provided, and is not important to the conclusions. This statement should be removed.

Thank you for this correction. We have rewritten this sentence accordingly (line 180 - 182).

Reviewer #2:

In this manuscript, authors report several new discoveries on the biology of MICU1. They show the localization of MICU1 at the mitochondrial inner boundary membrane (IBM) and describe domains responsible for such localization. They also show the importance of MICU1 in mitochondrial cristae width, an effect almost comparable to that of OPA1. At the end of the paper authors describe the importance of UCP2 and MICU1 methylation for MCU anchoring at the IBM. In its present form, the manuscript presents provocative and potentially important data. However, some experiments are required to support the conclusion of the paper.

We thank the referee for the very insightfully and constructive review and addressed each individual point as indicated below:

Our main concern lies on using mostly microscopy data without any biochemical data supporting them.

1) first outer membrane diameter over 500nm as shown in Fig.1, Fig.6 and 7 are double the size of what was measured by other groups using high resolution microscopy (Jakobs and Wurm 2014 in Current Opinion in Chemical Biology Volume 20, June 2014, Pages 9-15 describe outer membrane diameter to be 280nm (STED microscopy) and Plecitá-Hlavatá et al showed it to be 250nm by PALM microscopy (FASEB J. 2016 May;30(5):1941-57.), suggesting that perhaps they are stressed and acquire a doughnut like shape. Most of the mitochondria are becoming swollen (they have a hole in mitotracker signal) and for example mitochondria after histamine treatment seems to have higher diameter than untreated. This should be measured and stated in the manuscript.

This is an important issue. However, Jakobs and Wurm stated in the third paragraph that “the width of mitochondrial tubules is typically between 250 and 500 nm” (Jakobs und Wurm 2014). Moreover, depending on the method (FWHM, peak-to-peak analysis, minor and major measurements in thresholded images) and spatial resolution of the used system these values might differ additionally. Notably, Plecitá-Hlavatá et al. (Plečić-Hlavatá et al. 2016) used Hep-G2 cells and size and structure of mitochondria of different cell types may vary. Hence, they applied PALM microscopy to measure the mitochondrial diameter might provide other numbers because of its better spatial resolution of PALM compared with SIM used in our study. Since the mitochondrial morphology is, as mentioned by the referee, an important point regarding mitochondrial healthiness, we added measurements of the mitochondrial morphology of different cell lines with and without histamine treatment (Results lines 277 – 281, new **Supplementary Figure 28**). The mitochondrial morphology of cells treated with siRNA targeted against taffazin was added (Results lines 131 to 133, new **Supplementary Figure 11**). The mitochondrial morphology of cells expressing WT-MICU1-, MICU1-R455F-, MICU1-R455K-, and

MICU1- Δ EF-YFP was added (Results; lines 149 to 152, new **Supplementary Figure 13**). The mitochondrial morphology of cells expressing WT-MICU1-YFP or C-MICU1-YFP treated with control, OPA1 or nonCDS-MICU1 siRNA was added (Results; lines 161 to 163 & 171, new **Supplementary Figure 15**). The mitochondrial morphology of cells expressing UCP2-mCherry and WT-MICU1-, MICU1-R455F- or MICU1-R455K-YFP, treated with control or PRMT1 siRNA and stimulated with or without Histamine was added (Results lines 328 to 331, new **Supplementary Figure 35**).

2) Is there any reference or clear experiment confirming the transmembrane domain in MICU1? So far there are only papers claiming cardiolipin association but not clear transmembrane localization (Kramer et al, EMBO Rep. 2017 Aug;18(8):1397-1411.). because of this discrepancy, authors shall verify if cardiolipin deficiency (eg, tafazzin deletion) displaces MICU1

According to the comment of the referee, we changed our figure and text accordingly and exchanged TMD (transmembrane domain) by MTS (mitochondrial targeting sequence) in Figure 2a and the respective figure legend.

We appreciate this suggestion very much. To include the potential role of cardiolipin for the IBM localization of MICU1. Therefore, we performed MICU1-YFP translocation measurements and found a redistribution of MICU1 into the whole IMM under tafazzin siRNA treatment. Furthermore, a slight reduction in form factor could be observed under TAF knockdown; most probably due to reduction in the mitochondrial diameter. We added these results to the manuscript in the result section (lines 129 – 135), discussion (lines 376 - 378), and new **Supplementary Figure 11**.

3) Another problem lies in the exact localization of MICU1. Authors claim intermembrane space localization but there is another publication showing mitochondrial matrix localization (Hoffman et al 2013, Cell Reports, Volume 5, Issue 6, 26 December 2013, Pages 1576-1588). Therefore, I would suggest more biochemical experiments to support the authors' conclusion, like standard carbonate extraction and proteinase K treatment to reveal membrane localization.

Within the discussion of Hoffman *et al.* 2013, the localization of MICU1 is based on the MICU1-YFP efflux upon treatment with mastoparan (mitochondrial permeability transition induction (Yamamoto et al. 2014)) and alamethicin (voltage-dependent ion channel formation (Pieta et al. 2012)) (Hoffman et al. 2013). Hoffman et al. did not report a MICU1 efflux upon treatment with mastoparan, but mitochondrial depletion from MICU1 with Alamethicin (Hoffman et al. 2013). We strongly believe that MICU1 is membrane-bound in a membrane potential-dependent manner. This would explain the lack of influence of Mastoparan but the observed effect of Alamethicin, as it is potentially reducing the membrane potential leading to an efflux of MICU1. Furthermore, Marchi *et al.* just recently published data confirming the IBM localization of MICU1 using Proteinase K and carbon extraction assays (Marchi et al. 2019). This has been included in our text in lines 351 - 353.

4) According to the supplementary figures, all of the tested proteins are 60X overexpressed in comparison to wt levels of proteins. This is a major concern. Could authors come to same conclusions using for example immunocytochemistry against native MICU1? What if overexpression of MICU1 just leads to artifacts? Maybe MICU1 overexpression is widening mitochondria. Did authors compare thickness of mitochondria (or IBM) in controls and cells overexpressing MICU1? All of the differences between wt and overexpression should be discussed in the manuscript. For example, what about cristae morphology? Are cristae tightened in OE cells, if upon ablation they are widened?

Thank you very much for this important point. Unfortunately, immunocytochemistry against native MICU1 would be technically not possible because: (1), as stated in the text, MICU1 was localized using paraform-aldehyde/glutaraldehyde fixation within the cristae using electron microscopy by Lam et al. (Lam et al. 2015). Chemical fixation leads to a depolarization of the inner mitochondrial membrane that yields rearrangement of MICU1, which binds electrostatically with its poly-basic domain to membrane phospholipids (e.g. cardiolipin) (Kamer et al. 2017). (2) Antibodies targeting endogenous MICU1 with the proper affinity and selectivity needed for immunocytochemistry are not known to us.

To address the point raised by the referee, we tested whether the expression of MICU1-YFP has an effect on mitochondrial morphology. We could not find any difference in volume, surface, elongation or flatness. These results have been mentioned in the results section (lines 71 - 73) and in new **Supplementary Figure 1**.

5) In Fig.2 authors show a change in localization of MICU1 upon membrane potential depletion. However, this experiment is raising many questions. Is MICU1 localization changed because of proteolytic cleavage or of changes in oligomerization? This questions can be answered by SDS and BNPAGE electrophoresis.

We performed experiments using Western blotting to show no proteolytic degradation of MICU1 in HeLa cells even after 10 min of oligomycin A/antimycin A incubation. These results have been added in the results (lines 116 – 118) and in new **Supplementary Figure 10**.

6) How does ΔC -MICU1-YFP interact with WT MICU1, if the C-helix was deleted? Can authors show this on native PAGE?

We completely agree with the referee and also don't think that ΔC -MICU1-YFP is directly interacting with WT-MICU1. Accordingly, we think that endogenous MICU1 and OPA1 together are tightening and closing the CJ, thus, inhibiting ΔC -MICU1-YFP diffusion into the cristae lumen.

7) In the second paragraph on page 5 authors conclude that "The contribution of OPA1 on the integrity of the CJ was verified with transmission electron microscopy (TEM) that revealed a widening of the CJ by the knockdown of OPA1 (Fig. 4a,b&d) and, thus, explains the movement of the ΔC -mutant of MICU1 into the CM". How does OPA KD explain the movement of the ΔC -mutant of MICU1 into the CM? Could authors better explain their conclusion?

We are sorry for being not clear in this point. In the present version, we have rewritten this sentence to gain clarity (lines 175 – 176).

8) What is the physiological significance of having MICU1 in the IBM if MCU is in the cristae? What is the relationship between MICU1 in the IBM and mPTP? Can MICU1 at the IBM have any effect on mPTP opening?

Thank you for this important point. According to our present data, the localization of MICU1 at the IBM serves as a Ca^{2+} -dependent diffusion trap for MCU (lines 471 - 473, 510 – 512, **Figure 8**). Hence, MICU1, as far as we can tell, is participating in the control of the CJ as main result of this work. To further address this important point raised by this referee, we performed further experiments that

highlight the spatiotemporal correlation of sub-mitochondrial Ca^{2+} signals indicating a sequence of Ca^{2+} propagation from the cytosol into the IMS inducing MICU1 disassembly, its subsequent propagation further into the CL, and finally into the matrix (results section: lines 249 - 261; discussion: lines 431 – 433; new **Figure 5, panels F, G & H**; new **Supplementary Figure 23**).

Reviewer #3:

Gottschalk et al present data relating to distribution of MICU1 and a number of other factors mostly in dual colour live cell SIM imaging with some data from electron microscopy. Although the paper appears to show some interesting results, some of the presented data does not appear to support the conclusion they come to and their discussion makes following their reasoning difficult.

We thank the referee for the critical but fair report and have addressed each single point raised as indicated below:

Main issues:

Figure 1) and text "we estimated a distance of 22 ± 4 nm between the distributions of MICU1 and TOM22"

Is this results estimated or measured? Whichever, it is an extremely bold claim, given the data is a) in different colours and so subject to dispersion and chromatic aberration, and b) their image resolution, although not stated or measured is likely to be somewhere around 120-140 nm and they claim to have a accuracy of < 5 nm after making just 24 measurements.

Thank you for this important point. We fully understand the referee's concerns and performed additional experiments to address this issue:

(a) Regarding the first question, we imaged Tetraspec Beads to analyze potential chromatic abbreviations of our microscopic setup. Before analyzing any distortions, the common microscopic adjustments were done. Hence a calibration for the grating, PSF correction using the objective correction ring, and the dual-cam registration was performed. The dual-cam registration was made with the Tetraspec beads. The NIS-Elements registration tool corrects for translation, rotation and compression/stretching alterations. Afterwards, five different fields of view were imaged with our dual-cam setup. The point distributions of the beads were fitted to a Gaussian distribution, and the respective x and y coordinates were listed using the QuickPALM plugin for ImageJ. The vectors in between the two colors for each bead were measured. For each bead, the closest neighbor was identified, the bead distance measured and the difference of the chromatic abbreviation vectors determined. This represents a linear interpolation of the vector-field in between the two colors. The changes of this vector field in absolute numbers depend on the distance between measured points, and, thus, only a relative systematic error can be defined. The mean chromatic abbreviation along the measured distance is approximately 1 %. This calculation does not include the orientation of the line plot through the mitochondria during the actual measurement relative to the vector-field and thus represents a worst case scenario. Altogether these experiments indicate that in our setup the chromatic aberration have only a minor impact on the measurements conducted using mCherry-

TOM22 and MICU1-YFP. We mentioned this findings in lines 79 – 83 and added these measurements and a scheme illustrating the procedure in new **Supplementary Figure 4**.

(b) The resolution in this particular model is not limited to the spatial resolution of the microscope of 120 nm alone. The measurement itself is limited to the microscope's ability to resolve the distance between both sides of the mitochondrion, or its diameter. Otherwise, the accuracy of the fitting process matching the real data is the only limiting step for measurement accuracy. We assume a barrel-shape like mitochondrion with the OMM and IBM as two different barrel layers. By convolution of a barrel shape curve with the diameter of 350 nm (mean diameter of mitochondria analyzed) and a Gaussian function with a $\text{FWHM}_{x,y}$ of 120 nm (resolution in x,y) and a FWHM_z of 280 nm (resolution in z), a density distribution can be generated. By projecting the distribution on the x-axes, a double peak distribution with a peak to peak distance of approximately 350 nm is plotted. By fitting the double peak distributions of MICU1-YFP and mCherry-TOM22 line plots to a double Gaussian function, the peak-to-peak distance, or the diameter of the mitochondrion, for each distribution can be measured with very high subpixel accuracy. We explained these findings in lines 83 - 87 added a scheme explaining the methodology in the manuscript as new **Supplementary Figure 6**. We further added data to directly compare MICU1-YFP and mCherry-TOM22 peak-to-peak distances and a dot-plot of the delta peak-to-peak distances with detailed statistical information in lines 74 - 79 and as new **Supplementary Figure 3** to further support our measurement.

I would be particularly worried that Z plane misalignment would image different regions in the different colours leading to possibly systematic errors in such a small sample size. Even if the imaging system is perfectly aligned, dispersion with the live cell will affect the Z magnification and hence image alignment between channels with imaging depth.

Indeed, as measured in our new experiments (lines 83 -88; new **Supplementary Figure 5**) we have a Z-plane misalignment of 45 nm on average. We measured a mean diameter of the measured mitochondria of around 350 nm. Given a misalignment of 45 nm, the illumination density between two colors changes for the barrel shaped mitochondrial model used. Convoluting a barrel shape with a Gaussian function placed +/- 100 nm away from the focal center results in an inhomogeneous intensity distribution, illustrated in new **Supplementary Figure 5**. This indeed is changing the peak-to-peak distance of the x-axes projection. The change in peak-to-peak distance depends on the z-misalignment and the focal spot of the objective within the modeled mitochondrion (**Supplementary Figure 6**). A linear correlation of the delta peak-to-peak distance between two colors with the off-focus distance to the middle of the mitochondrion can be observed. We have to assume a normal distribution with regard to the off-focus relative to the mitochondrial center, as during the imaging process the mitochondria are focused, but not as accurate to the absolute center. A mean of approximately 1.6 nm peak-to-peak distance difference between two colors has to be taken as a systematic error of this particular measurement setup (new **Supplementary Figure 6**).

Figure 2b) The selection of the marked region seems to provide the answer that the authors want to find. Moving the selected line scan roughly 500 nm NW would produce a result similar to T=270 at both time points, whereas moving a similar distance SE would produce the T=0 result at both time points. The data selection seems extremely biased.

We agree with the referee as we definitely see a heterogeneous intra- and inter-mitochondrial rearrangement. To address these uncertainties, we always used multiple line profiles along the entire mitochondrion (as now correctly indicated by the boxes in the **Figure 2b**). Furthermore, we show in **Figure 2d** the quantification of MICU1 rearrangement from the IBM to the whole IMM using the IBM

association factor, which includes all mitochondria in the field of view and is thereby unbiased with regard to mitochondrial selection.

I am also surprised by the significance in plot 2g, however since you don't state what the box and whiskers represent on the plot it is impossible to tell if this significance is likely to be real or not.

Thank you very much for this important advice. We now added the general boxplot layout in the methods section of our manuscript (lines 786 – 787).

In supplementary figure 2, the IMM/MTG staining appears morphologically quite different in TOM22 compared to all the others. The mitochondria seems to be much thinner and less heavily stained for IMM/MTG

To address this point, we now added a diagram in **Supplementary Figure 7** (right panel) showing the MTG and mCherry FWHM (full width of half maximum) of mitochondria measured in HeLa cells stained with MTG and transfected with mCherry-TOM22, MCU-mCherry, EMRE-mCherry, or UCP2-mCherry. These data reveal that thickness of mitochondria does not differ between the various labels (i.e. mCherry-TOM22, MCU-mCherry, EMRE-mCherry, and UCP2-mCherry) used.

It seems that figure 3 has a similar shift in the morphology with much thinner mitochondria in the delC-MICU1 images whether there is a wildtype knockdown or not.

To address this important point, we have performed further experiments and measured mitochondrial morphology (form factor, aspect ratio, major/minor diameter, area). According to these measurements, the Δ C-MICU1 does affect the morphology of mitochondria compared to WT-MICU1 expression. These new findings are mentioned in the text (lines 161 - 163) and presented as new **Supplementary Figure 15**.

Supplementary Fig 7, movement is shown in arbitrary units, how is this sensible. Also how is this movement defined? Mean Square Displacement? $\mu\text{m/s}$?

We are sorry for this mistake and now have changed the axes description to "changed mitochondrial area per frame in %" as now shown in **Supplementary Figure 17**. It is a normalized measurement without dimension.

The text states "after stimulation with histamine, MCU joins MICU1 to exclusively localise in the IBM." This just isn't true. Yes it is more in the same places as the MICU1 but definitely not exclusively, look at the blown up plot in the bottom right plan of 6a, there are significant purple areas which are not green.

We weakened this term and replaced "exclusively" with "predominantly" (lines 267-268).

Fig 6b shows box and whisker plots for 4 conditions, with one being marked as significant, but what is this test against? I assume that it is the +hist control but you don't say this? How do you account for the lower IBM association index in the -hist experiment?

We thank the referee for the awareness. Indeed, the “ * ” for the indication of significance was shifted accidentally. We now corrected that. The lower IBM association index is likely due to the complex lack of binding of MCU to the Δ C-MICU1.

In discussing this figure you state "These data indicate...active MCUC dynamically assembles because MICU1 traps MCU in the IBM", I think this is a proposed mechanism suitable for the discussion but not in the results section. Also Supplementary fig 13 is not strong evidence for this, the IBM association factor goes up by only about 10%.

We have omitted this sentence as suggested by the referee.

We agree with the referee on the rather small change in the IBM association index. However, former Supplementary Figure 13 (now **Supplementary Figure 26**) shows the IBM association index of MCU-mCherry depending on the expression of untagged MICU1. Accordingly, we cannot be sure whether or not all cells imaged are indeed transfected with MICU1. That adds an error as cells not transfected with MICU1 are included in the measurements. Furthermore, in this particular case and in that shown in **Figure 8c** we had to use sequential dual-color imaging instead of simultaneous like in the other experiments, because of high fluorescent MTG and MCU-mCherry intensity differences. Sequential dual-color imaging leads to a time delay between images of 1 to 1.5 seconds, inducing an additional error for the calculation of the IBM association factor. Please note that both figures (i.e. **Supplementary Figure 26** and **Figure 8c**) have a reduced IBM association index, thus supporting our assumption.

Page 8: the references to sup fig 15,16 are wrong, and refer to data which isn't there, and sup fig 16 is missing.

corrected

Discussion. The discussion is hard to follow with a multitude of similar acronyms in quick succession (CJ,CM,CL) for those not intimately familiar with these it is very hard to follow the logic of what the authors are trying to describe. I think they also overstate their case in several places "localisation of MICU1 depends on the proteins ability of electrostatic repulsive power." All the authors show is that the 70-140 residue region is required.

As we measured the dependency of MICU1 IBM localization not only with the mutants but also in regard of membrane potential and cardiolipin levels (Kamer et al. 2017), the conclusive theory of electrostatic-based restriction of MICU1 to the IBM by the cardiolipin is justified in our eyes.

They also state that

"Our findings that knockdown of OPA1 did not change the exclusive IBM location of wild-type MICU1 but facilitated the redistribution of Δ C-MICU1 into the CM even in the presence of endogenous MICU1

suggest a model in which the sub-mitochondrial localization of MICU1 to the IBM is dependent on the stability of the CJ."

I don't follow the argument as to why the re-localisation is then dependant on CJ stability?

Our data clearly show that OPA1 and/or MICU1 depletion yields reduced stability of the CJ indicated by multiple methods demonstrating a reduced membrane potential, increased basal cristae Ca^{2+} , cytochrome C relocation, and the electron-microscopy data (increase CJ diameter). Under such condition, the ΔC -MICU1-YFP was found also in the CM. Accordingly, we believe that a discussion (!) on the importance of the integrity of the CJ for the distribution of proteins in the IMM (i.e. CM vs IBM) is justified.

Overall the discussion is extremely hard to follow and could benefit from a diagram showing how the proteins and CJ structure interact in their proposed mechanism.

With all respect, we do not understand this point. In fact, we provide several schemes in the manuscript (**Figure 1, Figure 4, Figure 5, Figure 8** as well as **Supplementary Figure 4, 5, 6, and 8**) to guide the reader through the manuscript. An overall scheme including all factors and interaction partners is hardly achievable and would be too speculative.

A large number of experiments depend upon the use of siRNA techniques, and yet limited no controls to demonstrate that the effects seen are not off target results. siRNA experiments need to have scrambled negative controls and two different siRNAs to be reliably interpreted. The one experiment which seems to have a negative control is shown in Sup fig 8, but there is no information about the "control si" used.

We thank the referee for raising this important point. Indeed, for all experiments conducted, randomized siRNA was used as control. We clarified this in the method section of the manuscript as well as in **Supplementary Table 1**.

Several of the image analyse techniques are said to include background subtraction, but how this is achieved is only stated in the Morphological analysis of mitochondria over time section, with no suggestion that this technique was used elsewhere.

Background was subtracted using an ImageJ plugin (Mosaic Suite, background subtractor, NIH) or a background ROI. Respective methods are indicated now in the methods section.

Many of the analysis routines use an Otsu auto threshold, whereas the morphological analysis uses a Yen auto threshold, why?

For the analysis process calculating the IBM association coefficient, the Otsu threshold was used because of its more stringent fragmentation characteristics. The Otsu threshold was taken due to "filling holes" and "dilatation and erosion" steps after thresholding. Otherwise, steps in the follow-up procedure might have been influenced negatively. Moreover, the Yen auto threshold was used for morphological analysis of mitochondria because of superior matching to the mitochondrial structures.

In any case, differences in both thresholding methods applied to our fluorescently labeled mitochondria are minor.

The extensive use of box and whisker plots is nice, however the authors must specify exactly what the boxes and whiskers represent, otherwise they are meaningless. There are also situations where it is not clear which values are being compared to test for statistical significance.

Box-plots are reported from minimum to maximum values as median, second and third quantile while whiskers cover the first and fourth quantile. This is stated now in the methods section (lines 786 – 787).

Minor issues.

Supplementary figure 5, it is hard to follow when mutants are called MICU1-R455F-YFP in the text and MICU1-F in the figure.

We thank the referee for the awareness and corrected the naming in the figures to MICU1-R455F-YFP or MICU1-R455K-YFP.

References:

Hoffman, Nicholas E.; Chandramoorthy, Harish C.; Shamugapriya, Santhanam; Zhang, Xueqian; Rajan, Sudarsan; Mallilankaraman, Karthik et al. (2013): MICU1 motifs define mitochondrial calcium uniporter binding and activity. In: *Cell reports* 5 (6), S. 1576–1588. DOI: 10.1016/j.celrep.2013.11.026.

Jakobs, Stefan; Wurm, Christian A. (2014): Super-resolution microscopy of mitochondria. In: *Current opinion in chemical biology* 20, S. 9–15. DOI: 10.1016/j.cbpa.2014.03.019.

Kamer, Kimberli J.; Grabarek, Zenon; Mootha, Vamsi K. (2017): High-affinity cooperative Ca²⁺ binding by MICU1-MICU2 serves as an on-off switch for the uniporter. In: *EMBO reports* 18 (8), S. 1397–1411. DOI: 10.15252/embr.201643748.

Lam, Stephanie S.; Martell, Jeffrey D.; Kamer, Kimberli J.; Deerinck, Thomas J.; Ellisman, Mark H.; Mootha, Vamsi K.; Ting, Alice Y. (2015): Directed evolution of APEX2 for electron microscopy and proximity labeling. In: *Nature methods* 12 (1), S. 51–54. DOI: 10.1038/nmeth.3179.

Marchi, Saverio; Corricelli, Mariangela; Branchini, Alessio; Vitto, Veronica Angela Maria; Missiroli, Sonia; Morciano, Giampaolo et al. (2019): Akt-mediated phosphorylation of MICU1 regulates mitochondrial Ca²⁺ levels and tumor growth. In: *The EMBO journal* 38 (2). DOI: 10.15252/embj.201899435.

Petrungaro, Carmelina; Zimmermann, Katharina M.; Küttner, Victoria; Fischer, Manuel; Dengjel, Jörn; Bogeski, Ivan; Riemer, Jan (2015): The Ca⁽²⁺⁾-Dependent Release of the Mia40-Induced MICU1-MICU2 Dimer from MCU Regulates Mitochondrial Ca(2+) Uptake. In: *Cell metabolism* 22 (4), S. 721–733. DOI: 10.1016/j.cmet.2015.08.019.

Phillips, Charles B.; Tsai, Chen-Wei; Tsai, Ming-Feng (2019): The conserved aspartate ring of MCU mediates MICU1 binding and regulation in the mitochondrial calcium uniporter complex. In: *eLife* 8. DOI: 10.7554/eLife.41112.

Pieta, Piotr; Mirza, Jeff; Lipkowski, Jacek (2012): Direct visualization of the alamethicin pore formed in a planar phospholipid matrix. In: *Proceedings of the National Academy of Sciences of the United States of America* 109 (52), S. 21223–21227. DOI: 10.1073/pnas.1201559110.

Plecitá-Hlavatá, Lydie; Engstová, Hana; Alán, Lukáš; Špaček, Tomáš; Dlasková, Andrea; Smolková, Katarína et al. (2016): Hypoxic HepG2 cell adaptation decreases ATP synthase dimers and ATP production in inflated cristae by mitofilin down-regulation concomitant to MICOS clustering. In: *FASEB journal : official publication of the Federation of American Societies for Experimental Biology* 30 (5), S. 1941–1957. DOI: 10.1096/fj.201500176.

Sancak, Yasemin; Markhard, Andrew L.; Kitami, Toshimori; Kovács-Bogdán, Erika; Kamer, Kimberli J.; Udeshi, Namrata D. et al. (2013): EMRE is an essential component of the mitochondrial calcium uniporter complex. In: *Science (New York, N.Y.)* 342 (6164), S. 1379–1382. DOI: 10.1126/science.1242993.

Tufi, Roberta; Gleeson, Tom; Stockum, Sophia von; Hewitt, Victoria; Lee, Juliette; Terriente-Felix, Ana et al. (2018): A comprehensive genetic characterisation of the mitochondrial Ca²⁺ uniporter in *Drosophila*. <http://dx.doi.org/10.1101/458174>

Yamamoto, Takenori; Ito, Mika; Kageyama, Keita; Kuwahara, Kana; Yamashita, Kikuji; Takiguchi, Yoshiharu et al. (2014): Mastoparan peptide causes mitochondrial permeability transition not by interacting with specific membrane proteins but by interacting with the phospholipid phase. In: *The FEBS journal* 281 (17), S. 3933–3944. DOI: 10.1111/febs.12930.

REVIEWERS' COMMENTS:

Reviewer #1 (Remarks to the Author):

The authors have largely addressed my concerns. This is an interesting manuscript that helps to reconcile some observations in the field and in particular helps to connect cristae junction dynamics to the uniporter complex.

Reviewer #2 (Remarks to the Author):

Authors satisfactorily addressed our concerns.

Reviewer #3 (Remarks to the Author):

Gottschalk et al present data relating to distribution of MICU1 and a number of other factors mostly in dual colour live cell SIM imaging with some data from electron microscopy.

The authors have addresses the majority of my comments form the original review. I still feel there is one outstanding issue.

I still don't feel that they adequately support their claim of measuring the inter protein distance between MICU1 in the IBM and TOM22 in the OMM.

They have gone to great lengths to calibrate the chromatic aberration in a bead sample. This will give good answers in XY and on the coverslip in Z, but they seem to have missed my point that dispersion will lead to linear scaling in Z. Measuring the Z focal shift of beads at the coverslip does not deal with this complication. I am not convinced that the small (~ 20 nm) size shift they see is not caused by a systematic focal shift with colour.

If they are focused on the centre of the mitochondria in the mCherry channel, then a consistent Z shift of 100-200 nm in the YFP channel could easily lead to a systematic error of 40 nm in the diameter and hence a measure separation of 20 nm. As the data appear to be 2D SIM stacks (there is no 3D SIM data presented in the paper) measuring Z shift in the actual samples appears to be impossible.

Additionally, in figure 1, they state it is 21 ± 13 nm, however the discussion says 22 ± 4 nm, which is true?

As this is the defining evidence that MICU1 is in the IBM I think this is a very significant point and needs to be shown to be true and not a systematic error.

Minor issues

line 345: should be "...MCU-Complex consists as a rather stable..."

line 348-349 The phrase " ...await consideration if one seeks to understand the molecular processes of mitochondrial Ca²⁺." Maybe more clearly phrased as "...should be considered in order to understand..."

line 361: ..." the comparison with TOM22 as a marker protein"

line 394 "...in cells that were depleted of wild-type"

line 407 "... a widening of the CJ has already been reported"

line 428 "..leads to increased oxygen"

line 453 "That indicates a strong discrepancy to our.." should probably read "Disagrees with"

Line 464: "...under control conditions, which is not..."

line 482 "...which then anchors EMRE."

Line 485: "..might reply on different regulatory mechanisms..."

line 555: "z-stacks of Tetraspek beads"

line 557 " ...For each bead the x and y abbreviation was calculated...". I'm not sure what word should be there but not abbreviation!

line 797: "All Box-plots show minimum to maximum values as median and second and third quantile, while whiskers cover the first and fourth quantile.". This is not entirely clear. I think you mean the central line is the median value, the box extends from 25% to 75% and the whiskers encompass all the data. Whatever it means this needs to be better worded.

Response to the referees:

We truly thank the referees very much for their kind and valuable comments. We feel that with their help and wise suggestions our manuscript was significantly improved. Below please find our final responses to the remaining points:

Reviewer #1 (Remarks to the Author):

The authors have largely addressed my concerns. This is an interesting manuscript that helps to reconcile some observations in the field and in particular helps to connect cristae junction dynamics to the uniporter complex.

Thank you

Reviewer #2 (Remarks to the Author):

Authors satisfactorily addressed our concerns.

Thank you

Referee #3: The authors have addresses the majority of my comments from the original review.

Thank you

I still feel there is one outstanding issue. I still don't feel that they adequately support their claim of measuring the inter protein distance between MICU1 in the IBM and TOM22 in the OMM.

They have gone to great lengths to calibrate the chromatic aberration in a bead sample. This will give good answers in XY and on the coverslip in Z, but they seem to have missed my point that dispersion will lead to linear scaling in Z. Measuring the Z focal shift of beads at the coverslip does not deal with this complication. I am not convinced that the small (~20 nm) size shift they see is not caused by a systematic focal shift with colour.

If they are focused on the centre of the mitochondria in the mCherry channel, then a consistent Z shift of 100-200 nm in the YFP channel could easily lead to a systematic error of 40 nm in the diameter and hence a measure separation of 20 nm. As the data appear to be 2D SIM stacks (there is no 3D SIM data presented in the paper) measuring Z shift in the actual samples appears to be impossible.

As this is the defining evidence that MICU1 is in the IBM I think this is a very significant point and needs to be shown to be true and not a systematic error.

This is an important point and we thank the referee who helps us to demonstrate the accuracy of our technique. According to the referee's criticism we can now demonstrate that the actual in cell measured z-axis dispersion within the two colors does not influence our result and does only introduce a minor systemic error. To address the referee's remaining issue, we have performed experiments in those we loaded HeLa cells with the inner membrane staining dyes mitotracker green and mitotracker red and evaluated the z-axis displacement within the two colors. As seen in the **attached figure**, that we suggest to include as new figure Suppl. Fig. 7 in the manuscript and to mention it in the Result Section

(lines 77 -79)(94-96) and in the Methods Section (lines 451-454)(626-629) , the z-shifts induced by dispersion in the intact cells only induces a minor systematic error that can be neglected for our conclusion.

(simple markup)(extended markup)

Furthermore, we included simultaneous measurements of MICU1-YFP and MICU1-mCherry as proper control of our previous experiments using MICU1-YFP and TOM22-mCherry. This new data are now shown as new figure 1d and mentioned in the Result Section (lines 71-72)(84-86) and Methods Section (line 542)(717).

(simple markup)(extended markup)

Additionally, in figure 1, they state it is 21+-13 nm, however the discussion says 22+-4nm, which is true?

We thank the referee for her/his awareness. We are sorry for this mistake. In answering the referees points from the first review we have performed further experiments that slightly changed the statistics. We have corrected this in the attached new version of this manuscript (line 283)(409).

(simple markup)(extended markup)

Minor issues:

All minor points have already been corrected.